# Analysis of Learning a Flow-based Generative Model from Limited Sample Complexity

**Hugo Cui**
Statistical Physics of Computation Laboratory
École Polytechnique Fédérale de Lausanne (EPFL)
Lausanne, Switzerland

**Florent Krzakala**
Information Learning and Physics Laboratory
École Polytechnique Fédérale de Lausanne (EPFL)
Lausanne, Switzerland

**Eric Vanden-Eijnden**
Courant Institute of Mathematical Science
New York University (NYU)
New York, USA

**Lenka Zdeborová**
Statistical Physics of Computation Laboratory
École Polytechnique Fédérale de Lausanne (EPFL)
Lausanne, Switzerland

## Abstract

We study the problem of training a flow-based generative model, parametrized by a two-layer autoencoder, to sample from a high-dimensional Gaussian mixture. We provide a sharp end-to-end analysis of the problem. First, we provide a tight closed-form characterization of the learnt velocity field, when parametrized by a shallow denoising auto-encoder trained on a finite number $n$ of samples from the target distribution. Building on this analysis, we provide a sharp description of the corresponding generative flow, which pushes the base Gaussian density forward to an approximation of the target density. In particular, we provide closed-form formulae for the distance between the means of the generated mixture and the mean of the target mixture, which we show decays as $\Theta_n(1/n)$. Finally, this rate is shown to be in fact Bayes-optimal.

Flow and diffusion-based generative models have introduced a shift in paradigm for density estimation and sampling problems, leading to state-of-the art algorithms e.g. in image generation (Rombach et al., 2022; Ramesh et al., 2022; Saharia et al., 2022). Instrumental in these advances was the realization that the sampling problem could be recast as a transport process from a simple –typically Gaussian– base distribution to the target density. Furthermore, the velocity field governing the flow can be characterized as the minimizer of a quadratic loss function, which can be estimated from data by (a) approximating the loss by its empirical estimate using available training data and (b) parametrizing the velocity field using a denoiser neural network. These ideas have been fruitfully implemented as part of a number of frameworks, including score-based diffusion models (Song & Ermon, 2019; Song et al., 2020; Karras et al., 2022; Ho et al., 2020), and stochastic interpolation (Albergo & Vanden-Eijnden, 2022; Albergo et al., 2023; Lipman et al., 2022; Liu et al., 2022). A tight analytical understanding of the learning of generative models from limited data, and the resulting generative process, is however still largely missing. This constitutes the research question addressed in the present manuscript.

A line of recent analytical works (Benton et al., 2023; Chen et al., 2022; 2023a;c;d; Wibisono & Yang, 2022; Lee et al., 2022; 2023; Li et al., 2023; De Bortoli et al., 2021; De Bortoli, 2022; Pidstrigach, 2022; Block et al., 2020) have mainly focused on the study of the transport problem, and provide rigorous convergence guarantees, taking as a starting point the assumption of an $L^2-$accurate estimate of the velocity or score. They hence bypass the investigation of the learning problem –and in particular the question of ascertaining the sample complexity needed to obtain such an accurate estimate. More importantly, the study of the effect of learning from a *limited* sample complexity (and thus e.g. of possible network overfitting and memorization) on the generated density, is furthermore left unaddressed. On the other hand, very recent works (Cui & Zdeborová, 2023; Shah et al., 2023) have characterized the learning of Denoising Auto-Encoders (DAEs) (Vincent et al., 2010; Vincent, 2011) in high dimensions on Gaussian mixture densities. Neither work however studies the consequences on the generative process. Bridging that gap, recent works have offered a *joint* analysis of the learning and generative processes. Oko et al. (2023); Chen et al. (2023b); Yuan

et al. (2023) derive rigorous bounds at finite sample complexity, under the assumption of data with a *low-dimensional* structure. Closer to our manuscript, a concurrent work (Mei & Wu, 2023) bounds the Kullback-Leibler distance between the generated and target densities, when parametrizing the flow using a ResNet, for high-dimensional graphical models. On the other hand, these bounds do not go to zero as the sample complexity increases, and are a priori not tight.

The present manuscript aims at complementing and furthering this last body of works, by providing a tight end-to-end analysis of a flow-based generative model – starting from the study of the high-dimensional learning problem with a finite number of samples, and subsequently elucidating the implications thereof on the generative process.

**Main contributions–** We study the problem of estimating and sampling a Gaussian mixture using a flow-based generative model, in the framework of stochastic interpolation (Albergo & Vanden-Eijnden, 2022; Albergo et al., 2023; Lipman et al., 2022; Liu et al., 2022). We consider the case where a non-linear two-layer DAE with one hidden unit is used to parametrize the velocity field of the associated flow, and is trained with a finite training set. In the high-dimensional limit,

• We provide a sharp asymptotic closed-form characterization of the learnt velocity field, as a function of the target Gaussian mixture parameters, the stochastic interpolation schedule, and the number of training samples $n$.
• We characterize the associated flow by providing a tight characterization of a small number of summary statistics, tracking the dynamics of a sample from the Gaussian base distribution as it is transported by the learnt velocity field.
• We show that even with a finite number of training samples, the learnt generative model allows to sample from a mixture whose mean asymptotically approaches the mean of the target mixture as $\Theta_n(1/n)$ in squared distance, with this rate being tight.
• Finally, we show that this rate is in fact Bayes-optimal.

The code used in the present manuscript is provided in this repository.

RELATED WORKS

**Diffusion and flow-based generative models** Score-based diffusion models (Song & Ermon, 2019; Song et al., 2020; Karras et al., 2022; Ho et al., 2020) build on the idea that any density can be mapped to a Gaussian density by degrading samples through an Ornstein-Uhlenbeck process. Sampling from the original density can then be carried out by time-reversing the corresponding stochastic transport, provided the score is known – or estimated. These ideas were subsequently refined in (Albergo & Vanden-Eijnden, 2022; Albergo et al., 2023; Lipman et al., 2022; Liu et al., 2022), which provide a flexible framework to bridge between two arbitrary densities in finite time.

**Convergence bounds** In the wake of the practical successes of flow and diffusion-based generative models, significant theoretical effort has been devoted to studying the convergence of such methods, by bounding appropriate distances between the generated and the target densities. A common assumption of (Benton et al., 2023; Chen et al., 2022; 2023a;c;d; Wibisono & Yang, 2022; Lee et al., 2022; 2023; Li et al., 2023; De Bortoli et al., 2021; De Bortoli, 2022; Pidstrigach, 2022; Block et al., 2020) is the availability of a good estimate for the score, i.e. an estimate whose average (population) squared distance with the true score is bounded by a small constant $\epsilon$. Under this assumption, Chen et al. (2022); Lee et al. (2022) obtain rigorous control on the Wasserstein and total variation distances with very mild assumptions on the target density. Ghio et al. (2023) explore the connections between algorithmic hardness of the score/flow approximation and the hardness of sampling in a number of graphical models.

**Asymptotics for DAE learning** The backbone of flow and diffusion-based generative models is the parametrization of the score or velocity by a denoiser-type network, whose most standard realization is arguably the DAE (Vincent et al., 2010; Vincent, 2011). Very recent works have provided a detailed analysis of its learning on denoising tasks, for data sampled from Gaussian mixtures. Cui & Zdeborová (2023) sharply characterize how a DAE can learn the mixture parameters with $n = \Theta_d(d)$ training samples when the cluster separation is $\Theta_d(1)$. Closer to our work, for arbitrary cluster separation, Shah et al. (2023) rigorously show that a DAE trained with gradient descent on the denoising diffusion probabilistic model loss (Ho et al., 2020) can recover the cluster means with a polynomial number of samples. While these works complement the aforediscussed

convergence studies in that they analyze the effect of a finite number of samples, neither explores the flow associated to the learnt score.

**Network-parametrized models** Tying together these two body of works, a very recent line of research has addressed the problem of bounding, at finite sample complexity, appropriate distances between the generated and target densities, assuming a network-based parametrization. Oko et al. (2023) provide such bounds when parametrizing the score using a class of ReLU networks. These bounds however suffer from the curse of dimensionality. Oko et al. (2023); Yuan et al. (2023); Chen et al. (2023b) surmount this hurdle by assuming a target density with low-dimensional structure. On a heuristic level, Biroli & Mézard (2023) estimate the order of magnitude of the sample complexity needed to sample from a high-dimensional Curie-Weiss model. Finally, a work concurrent to ours (Mei & Wu, 2023) derives rigorous bounds for a number of high-dimensional graphical models. On the other hand, these bounds are a priori not tight, and do not go to zero as the sample complexity becomes large. The present manuscript aims at furthering this line of work, and provides a *sharp* analysis of a high-dimensional flow-based generative model.

# 1 SETTING

We start by giving a concise overview of the problem of sampling from a target density $\rho_1$ over $\mathbb{R}^d$ in the framework of stochastic interpolation (Albergo & Vanden-Eijnden, 2022; Albergo et al., 2023).

**Recasting sampling as an optimization problem** Samples from $\rho_1$ can be generated by drawing a sample from an easy-to-sample base density $\rho_0$ –henceforth taken to be a standard Gaussian density $\rho_0 = \mathcal{N}(0, \mathbb{I}_d)$–, and evolving it according to the flow described by the ordinary differential equation (ODE)

$$\frac{d}{dt} \boldsymbol{X}_t = \boldsymbol{b}(\boldsymbol{X}_t, t), \tag{1}$$

for $t \in [0, 1]$. Specifically, as shown in Albergo et al. (2023), if $\boldsymbol{X}_{t=0} \sim \rho_0$, then the final sample $\boldsymbol{X}_{t=1}$ has probability density $\rho_1$, if the velocity field $\boldsymbol{b}(\boldsymbol{x}, t)$ governing the flow (1) is given by

$$\boldsymbol{b}(\boldsymbol{x}, t) = \mathbb{E}[\dot{\alpha}(t)\boldsymbol{x}_0 + \dot{\beta}(t)\boldsymbol{x}_1 | \boldsymbol{x}_t = \boldsymbol{x}], \tag{2}$$

where we denoted $\boldsymbol{x}_t \equiv \alpha(t)\boldsymbol{x}_0 + \beta(t)\boldsymbol{x}_1$ and the conditional expectation bears over $\boldsymbol{x}_1 \sim \rho_1$, $\boldsymbol{x}_0 \sim \rho_0$, with $\boldsymbol{x}_0 \perp \boldsymbol{x}_1$. The result holds for any fixed choice of schedule functions $\alpha, \beta \in \mathcal{C}^2([0, 1])$ satisfying $\alpha(0) = \beta(1) = 1, \alpha(1) = \beta(0) = 0$, and $\alpha(t)^2 + \beta(t)^2 > 0$ for all $t \in [0, 1]$. In addition to the velocity field $\boldsymbol{b}(\boldsymbol{x}, t)$, it is convenient to consider the field $\boldsymbol{f}(\boldsymbol{x}, t)$, related to $\boldsymbol{b}(\boldsymbol{x}, t)$ by the simple relation

$$\boldsymbol{b}(\boldsymbol{x}, t) = \left( \dot{\beta}(t) - \frac{\dot{\alpha}(t)}{\alpha(t)} \beta(t) \right) \boldsymbol{f}(\boldsymbol{x}, t) + \frac{\dot{\alpha}(t)}{\alpha(t)} \boldsymbol{x}. \tag{3}$$

Note that $\boldsymbol{f}(\boldsymbol{x}, t)$ can be alternatively expressed as $\mathbb{E}[\boldsymbol{x}_1 | \boldsymbol{x}_t = \boldsymbol{x}]$, and thus admits a natural interpretation as a *denoising* function, tasked with recovering the target value $\boldsymbol{x}_1$ from the interpolated (noisy) sample $\boldsymbol{x}_t$. The denoiser $\boldsymbol{f}(\boldsymbol{x}, t)$ can furthermore characterized as the minimizer of the objective

$$\mathcal{R}[\boldsymbol{f}] = \int\limits_0^1 \mathbb{E} \left\| \boldsymbol{f}(\boldsymbol{x}_t, t) - \boldsymbol{x}_1 \right\|^2 dt. \tag{4}$$

The loss (4) is a simple sequence of quadractic *denoising* objectives.

**Learning the velocity from data** There are several technical hurdles in carrying out the minimization (4). First, since the analytical form of $\rho_1$ is generically unknown, the population risk has to be approximated by its empirical version, provided a dataset $\mathcal{D} = \{\boldsymbol{x}_1^\mu, \boldsymbol{x}_0^\mu\}_{\mu=1}^n$ of $n$ training samples $\boldsymbol{x}_1^\mu$ ($\boldsymbol{x}_0^\mu$) independently drawn from $\rho_1$ ($\rho_0$) is available. Second, the minimization in (4) bears over a time-dependent vector field $\boldsymbol{f}$. To make the optimization tractable, the latter can be parametrized at each time step $t$ by a separate neural network $\boldsymbol{f}_{\theta_t}(\cdot)$ with trainable parameters $\theta_t$. Under those approximations, the population risk (4) thus becomes

$$\hat{\mathcal{R}}(\{\theta_t\}_{t \in [0,1]}) = \int\limits_0^1 \sum_{\mu=1}^n \left\| \boldsymbol{f}_{\theta_t}(\boldsymbol{x}_t^\mu) - \boldsymbol{x}_1^\mu \right\|^2 dt. \tag{5}$$

Remark that in practice, the time $t$ can enter as an input of the neural network, and only one network then needs to be trained. In the present manuscript however, for technical reasons, we instead consider the case where a *separate* network is trained *for each time step* $t$. Besides, note that since the base density $\rho_0$ is a priori easy to sample from, one could in theory augment the dataset $\mathcal{D}$ with several samples from $\rho_0$ for each available $\boldsymbol{x}_1^\mu$. For conciseness, we do not examine such an augmentation technique in the present manuscript, and leave a precise investigation thereof to future work. Denoting by $\{\hat{\theta}_t\}_{t\in[0,1]}$ the minimizer of (5), the learnt velocity field $\hat{\boldsymbol{b}}$ is related to the trained denoiser $\boldsymbol{f}_{\hat{\theta}_t}$ by (4) as

$$\hat{\boldsymbol{b}}(\boldsymbol{x},t) = \left( \dot{\beta}(t) - \frac{\dot{\alpha}(t)}{\alpha(t)}\beta(t) \right) \boldsymbol{f}_{\hat{\theta}_t}(\boldsymbol{x}) + \frac{\dot{\alpha}(t)}{\alpha(t)}\boldsymbol{x}. \tag{6}$$

The sampling can finally be carried out by using $\hat{\boldsymbol{b}}$ as a proxy for the unknown $\boldsymbol{b}$ in (1):

$$\frac{d}{dt}\boldsymbol{X}_t = \hat{\boldsymbol{b}}(\boldsymbol{X}_t,t) \tag{7}$$

Note that the solution $\boldsymbol{X}_1$ at time $t = 1$ of the ODE (7) has a law $\hat{\rho}_1 \neq \rho_1$ due to the two approximations in going from the population function-space objective (4) to the empirical parametric proxy (5). The present manuscript presents a sharp analysis of the learning problem (5) and the resulting flow (7) for a solvable model, which we detail below.

**Data model** We consider the case of a target density $\rho_1$ given by a binary isotropic and homoscedastic Gaussian mixture

$$\rho_1 = \frac{1}{2}\mathcal{N}(\boldsymbol{\mu}, \sigma^2\mathbb{I}_d) + \frac{1}{2}\mathcal{N}(-\boldsymbol{\mu}, \sigma^2\mathbb{I}_d). \tag{8}$$

Each cluster is thus centered around its mean $\pm\boldsymbol{\mu}$ and has variance $\sigma^2$. For definiteness, we consider here a balanced mixture, where the two clusters have equal relative probabilities, and defer the discussion of the imbalanced case to Appendix D. Note that a sample $\boldsymbol{x}_1^\mu$ can then be decomposed as $\boldsymbol{x}_1^\mu = s^\mu\boldsymbol{\mu} + \boldsymbol{z}^\mu$, with $s^\mu \sim \mathcal{U}(\{-1,+1\})$ and $\boldsymbol{z}^\mu \sim \mathcal{N}(0, \sigma^2\mathbb{I}_d)$. Finally, note that the closed-form expression for the exact velocity field $\boldsymbol{b}$ (1) associated to the density $\rho_1$ is actually known (see e.g. Efron (2011); Albergo et al. (2023)). This manuscript explores the question whether a neural network can learn a good approximate $\hat{\boldsymbol{b}}$ thereof *without* any knowledge of the density $\rho_1$, and only from a finite number of samples drawn therefrom.

**Network architecture** We consider the case where the denoising function $\boldsymbol{f}$ (4) is parametrized with a two-layer non-linear DAE with one hidden neuron, and –taking inspiration from modern practical architectures such as U-nets (Ronneberger et al., 2015)– a trainable skip connection:

$$\boldsymbol{f}_{\boldsymbol{w}_t,c_t}(\boldsymbol{x}) = c_t \times \boldsymbol{x} + \boldsymbol{w}_t \times \varphi(\boldsymbol{w}_t^\top\boldsymbol{x}), \tag{9}$$

where $\varphi$ is assumed to tend to 1 (resp. $-1$) as its argument tends to $+\infty$ (resp $-\infty$). Sign, tanh and erf are simple examples of such an activation function. The trainable parameters are therefore $c_t \in \mathbb{R}, \boldsymbol{w}_t \in \mathbb{R}^d$. Note that (9) is a special case of the architecture studied in Cui & Zdeborová (2023). It differs from the very similar network considered in Shah et al. (2023) in that it covers a slightly broader range of activation functions (Shah et al. (2023) address the case $\varphi = \tanh$), and in that the skip connection istrainable –rather than fixed–. Since we consider the case where a separate network is trained at every time step, the empirical risk (5) decouples over the time index $t$. The parameters $\boldsymbol{w}_t, c_t$ of the DAE (9) should therefore minimize

$$\hat{\mathcal{R}}_t(\boldsymbol{w}_t, c_t) = \sum_{\mu=1}^n \|\boldsymbol{f}_{c_t,\boldsymbol{w}_t}(\boldsymbol{x}_t^\mu) - \boldsymbol{x}_1^\mu\|^2 + \frac{\lambda}{2}\|\boldsymbol{w}_t\|^2, \tag{10}$$

where for generality we also allowed for the presence of a $\ell_2$ regularization of strength $\lambda$. We remind that $\boldsymbol{x}_t^\mu = \alpha(t)\boldsymbol{x}_0^\mu + \beta(t)\boldsymbol{x}_1^\mu$, with $\{\boldsymbol{x}_1^\mu\}_{\mu=1}^n$ (resp. $\{\boldsymbol{x}_0^\mu\}_{\mu=1}^n$) $n$ training samples independently drawn from the target density $\rho_1$ (8) (resp. the base density $\rho_0 = \mathcal{N}(0, \mathbb{I}_d)$), collected in the training set $\mathcal{D}$.

**Asymptotic limit** We consider in this manuscript the asymptotic limit $d \to \infty$, with $n, \|\boldsymbol{\mu}\|^2/d, \sigma = \Theta_d(1)$. For definiteness, in the following, we set $\|\boldsymbol{\mu}\|^2/d = 1$. Note that Cui & Zdeborová (2023) consider the different limit $\|\boldsymbol{\mu}\| = \Theta_d(1)$. Shah et al. (2023) on the other hand address a larger range of asymptotic limits, including the present one, but does not provide tight characterizations, nor an analysis of the generative process.

## 2 LEARNING

In this section, we first provide sharp closed-form characterizations of the minimizers $\hat{c}_t, \hat{w}_t$ of the objective $\hat{\mathcal{R}}_t$ (10). The next section discusses how these formulae can be leveraged to access a tight characterization of the associated flow.

**Result 2.1.** **(Sharp characterization of minimizers of (10))** *For any given activation $\varphi$ satisfying $\varphi(x) \xrightarrow{x \to \pm\infty} \pm 1$ and any $t \in [0, 1]$, in the limit $d \to \infty$, $n, \|\boldsymbol{\mu}\|^2/d, \sigma = \Theta_d(1)$, the skip connection strength $\hat{c}_t$ minimizing (10) is given by*

$$\hat{c}_t = \frac{\beta(t)(\lambda(1+\sigma^2) + (n-1)\sigma^2)}{\alpha(t)^2(\lambda+n-1) + \beta(t)^2(\lambda(1+\sigma^2) + (n-1)\sigma^2)}. \tag{11}$$

*Furthermore, the learnt weight vector $\hat{w}_t$ is asymptotically contained in $\mathrm{span}(\boldsymbol{\mu}_{\mathrm{emp.}}, \boldsymbol{\xi})$ (in the sense that its projection on the orthogonal space $\mathrm{span}(\boldsymbol{\mu}_{\mathrm{emp.}}, \boldsymbol{\xi})$ has asymptotically vanishing norm), where*

$$\boldsymbol{\xi} \equiv \sum_{\mu=1}^{n} s^\mu x_0^\mu, \qquad\qquad \boldsymbol{\mu}_{\mathrm{emp.}} = \frac{1}{n}\sum_{\mu=1}^{n} s^\mu x_1^\mu. \tag{12}$$

*In other words, $\boldsymbol{\mu}_{\mathrm{emp.}}$ is the empirical mean of the training samples. We remind that $s^\mu = \pm 1$ was defined below (8) and indicates the cluster the $\mu-$th sample $x_1^\mu$ belongs to. The components of $\hat{w}_t$ along each of these three vectors is described by the summary statistics*

$$m_t = \frac{\boldsymbol{\mu}_{\mathrm{emp.}}^\top \hat{w}_t}{d(1+\sigma^2/n)}, \qquad\qquad q_t^\xi = \frac{\hat{w}_t^\top \boldsymbol{\xi}}{nd}, \tag{13}$$

*which concentrate as $d \to \infty$ to the quantities characterized by the closed-form formulae*

$$\begin{cases} m_t = \frac{n}{\lambda+n} \frac{\alpha(t)^2(\lambda+n-1)}{\alpha(t)^2(\lambda+n-1)+\beta(t)^2(\lambda(1+\sigma^2)+(n-1)\sigma^2)} \\ q_t^\xi = \frac{-\alpha(t)}{\lambda+n} \frac{\beta(t)(\lambda(1+\sigma^2)+(n-1)\sigma^2)}{\alpha(t)^2(\lambda+n-1)+\beta(t)^2(\lambda(1+\sigma^2)+(n-1)\sigma^2)} \end{cases}. \tag{14}$$

The derivation of Result 2.1 is detailed in Appendix A, and involves a heuristic partition function computation, borrowing ideas from statistical physics. The theoretical predictions for the skip connection strength $\hat{c}_t$ and the component $m_t, q_t^\xi$ of the weight vector $\hat{w}_t$ are plotted as solid lines in Fig. 1, and display good agreement with numerical simulations, corresponding to training the DAE (9) on the risk (10) using the `Pytorch` (Paszke et al., 2019) implementation of the Adam optimizer (Kingma & Ba, 2014).

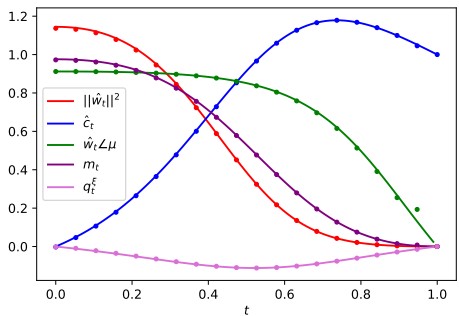

A notable consequence of (13) is that the weight vector $\hat{w}_t$ is contained at all times $t$ in the two-dimensional subspace spanned by the empirical cluster mean $\boldsymbol{\mu}_{\mathrm{emp.}}$ and the vectors $\boldsymbol{\xi}$ (12) – in other words, the learnt weights align to some extent with the empirical mean, but still possess a non-zero component along $\boldsymbol{\xi}$, which is orthogonal thereto. $\boldsymbol{\xi}$ subsumes the aggregated effect of the base vectors $\{x_0^\mu\}_{\mu=1}^n$ used in the train set. Rather remarkably, the training samples thus only enter in the characterization of $\hat{w}_t$ through the form of simple sums (12). Since the vector $\boldsymbol{\xi}$ is associated to the training samples, the fact that the learnt vector $\hat{w}_t$ has non-zero components along $\boldsymbol{\xi}$ hence signals a form of overfitting and memorization. Interestingly, Fig. 1 shows that the extent of this overfitting is non-monotonic

Figure 1: $n = 4, \sigma = 0.9, \lambda = 0.1, \alpha(t) = 1 - t, \beta(t) = t, \varphi = \tanh$. Solid lines: theoretical predictions of Result 2.1: squared norm of the DAE weight vector $\|\hat{w}_t\|^2$ (red), skip connection strength $\hat{c}_t$ (blue) cosine similarity between the weight vector $\hat{w}_t$ and the target cluster mean $\boldsymbol{\mu}, \hat{w}_t \angle \mu \equiv \hat{w}_t^\top \boldsymbol{\mu}/\|\boldsymbol{\mu}\|\|\hat{w}_t\|$ (green), components $m_t, q_t^\xi$ of $\hat{w}_t$ along the vectors $\boldsymbol{\mu}_{\mathrm{emp.}}, \boldsymbol{\xi}$ (purple, pink, orange). Dots: numerical simulations in dimension $d = 5 \times 10^4$, corresponding to training the DAE (9) on the risk (10) using the `Pytorch` implementation of full-batch Adam, with learning rate 0.0001 over $4 \times 10^4$ epochs and weight decay $\lambda = 0.1$. The experimental points correspond to a single instance of the model.

in time, as $|q_t^\xi|$ first increases then decreases. Finally, note that this effect is as expected mitigated as the number of training samples $n$ increases. From (14), for large $n$, $m_t = \Theta_n(1)$ while the components $q_t^\xi$ is suppressed as $\Theta_n(1/n)$. These scalings are further elaborated upon in Remark B.3 in Appendix B. Finally, Result 2.1 and equation (6) can be straightforwardly combined to yield a sharp characterization of the learnt estimate $\hat{b}$ of the velocity field $b$ (1). This characterization can be in turn leveraged to build a tight description of the generative flow (7). This is the object of the following section.

## 3 GENERATIVE PROCESS

While Corollary 2.1, together with the definition (6), provides a concise characterization of the velocity field $\hat{b}$, the sampling problem (7) remains formulated as a high-dimensional, and therefore hard to analyze, transport process. The following result shows that the dynamics of a sample $X_t$ following the differential equation (7) can nevertheless be succinctly tracked using a finite number of scalar summary statistics.

**Result 3.1.** (*Summary statistics*) *Let $X_t$ be a solution of the ordinary differential equation (7) with initial condition $X_0$. For a given $t$, the projection of $X_t$ on $\mathrm{span}(\boldsymbol{\mu}_{\mathrm{emp.}}, \boldsymbol{\xi}$ is characterized by the summary statistics*

$$M_t \equiv \frac{X_t^\top \boldsymbol{\mu}_{\mathrm{emp.}}}{d(1+\sigma^2/n)}, \qquad\qquad Q_t^\xi \equiv \frac{X_t^\top \boldsymbol{\xi}}{nd}. \qquad (15)$$

*With probability asymptotically $1/2$ the summary statistics $M_t, Q_t^\xi$ (15) concentrate for all $t$ to the solution of the ordinary differential equations*

$$\begin{cases} \frac{d}{dt} M_t = \left( \dot{\beta}(t)\hat{c}_t + \frac{\dot{\alpha}(t)}{\alpha(t)}(1-\hat{c}_t\beta(t)) \right) M_t + \left( \dot{\beta}(t) - \frac{\dot{\alpha}(t)}{\alpha(t)}\beta(t) \right) m_t \\ \frac{d}{dt} Q_t^\xi = \left( \dot{\beta}(t)\hat{c}_t + \frac{\dot{\alpha}(t)}{\alpha(t)}(1-\hat{c}_t\beta(t)) \right) Q_t^\xi + \left( \dot{\beta}(t) - \frac{\dot{\alpha}(t)}{\alpha(t)}\beta(t) \right) q_t^\xi \end{cases}, \qquad (16)$$

*with initial condition $M_0 = Q_0^\xi = 0$, and with probability asymptotically $1/2$ they concentrate to minus the solution of (16). Furthermore, the orthogonal component $X_t^\perp \in \mathrm{span}(\boldsymbol{\mu}_{\mathrm{emp.}}, \boldsymbol{\xi})^\perp$ obeys the simple linear differential equation*

$$\frac{d}{dt} X_t^\perp = \left( \dot{\beta}(t)\hat{c}_t + \frac{\dot{\alpha}(t)}{\alpha(t)}(1-\hat{c}_t\beta(t)) \right) X_t^\perp. \qquad (17)$$

*Finally, the statistic $Q_t \equiv \|X_t\|^2/d$ is given with high probability by*

$$Q_t = M_t^2(1+\sigma^2/n) + n(Q_t^\xi)^2 + e^{2\int_0^t \left( \dot{\beta}(t)\hat{c}_t + \frac{\dot{\alpha}(t)}{\alpha(t)}(1-\hat{c}_t\beta(t)) \right)dt}. \qquad (18)$$

A heuristic derivation of Result 3.1 is provided in Appendix B. Taking a closer look at (16), it might seem at first from equations (16) that there is a singularity for $t = 1$ since $\alpha(1) = 0$ in the denominator. Remark however that both $1 - \beta(t)\hat{c}_t$ (11) and $m_t$ (14) are actually proportional to $\alpha(t)^2$, and therefore (16) is in fact also well defined for $t = 1$. In practice, the numerical implementation of a generative flow like (7) often involves a discretization thereof, given a discretization scheme $\{t_k\}_{k=0}^N$ of $[0,1]$, where $t_0 = 0$ and $t_N = 1$:

$$X_{t_{k+1}} = X_{t_k} + \hat{b}(X_{t_k}, t_k)(t_{k+1} - t_k). \qquad (19)$$

The evolution of the summary statistics introduced in Result 3.1 can be rephrased in more actionable form to track the discretized flow (19).

**Remark 3.2.** (*Summary statistics for the discrete flow*) *Let $\{X_{t_k}\}_{k=0}^N$ be a solution of the discretized learnt flow (7), for an arbitrary discretization scheme $\{t_k\}_{k=0}^N$ of $[0,1]$, where $t_0 = 0$ and $t_N = 1$, with initial condition $X_{t_0} \sim \rho_0$. The summary statistics introduced in Result 3.1 are then equal to the solutions of the recursions*

$$\begin{cases} M_{t_{k+1}} = M_{t_k} + \delta t_k \left( \dot{\beta}(t_k)\hat{c}_{t_k} + \frac{\dot{\alpha}(t_k)}{\alpha(t_k)}(1-\hat{c}_{t_k}\beta(t_k)) \right) M_{t_k} + \delta t_k \left( \dot{\beta}(t_k) - \frac{\dot{\alpha}(t_k)}{\alpha(t_k)}\beta(t_k) \right) m_{t_k} \\ Q_{t_{k+1}}^\xi = Q_{t_k}^\xi + \delta t_k \left( \dot{\beta}(t_k)\hat{c}_{t_k} + \frac{\dot{\alpha}(t_k)}{\alpha(t_k)}(1-\hat{c}_{t_k}\beta(t_k)) \right) Q_{t_k}^\xi + \delta t_k \left( \dot{\beta}(t_k) - \frac{\dot{\alpha}(t_k)}{\alpha(t_k)}\beta(t_k) \right) q_{t_k}^\xi \end{cases}, \qquad (20)$$

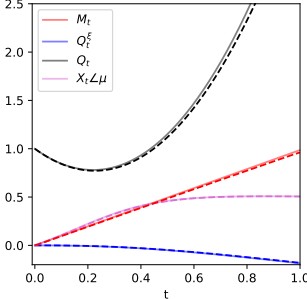 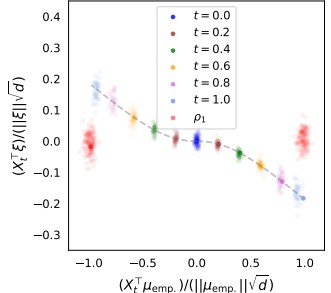 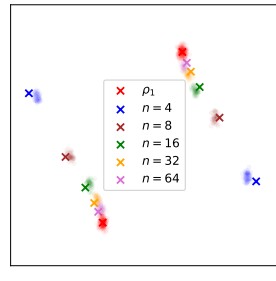

Figure 2: In all three plots, $\lambda = 0.1, \alpha(t) = 1 - t, \beta(t) = t, \varphi = \text{sign}$. (**left**) $\sigma = 1.5, n = 8$. Temporal evolution of the summary statistics $M_t, Q_t^\xi, Q_t, \boldsymbol{X}_t \angle \boldsymbol{\mu}$ (15). Solid lines correspond to the theoretical prediction of (15) in Result 3.1, while dashed lines correspond to numerical simulations of the generative model, by discretizing the differential equation (7) with step size $\delta t = 0.01$, and training a separate DAE for each time step using Adam with learning rate 0.01 for 2000 epochs. All experiments were conducted in dimension $d = 5000$, and a single run is represented. (**middle**) $\sigma = 2, n = 16$. Projection of the distribution of $\boldsymbol{X}_t$ (7) in $\text{span}(\boldsymbol{\mu}_{\text{emp.}}, \boldsymbol{\xi})$, transported by the velocity field $\hat{\boldsymbol{b}}$ (6) learnt from data. The point clouds correspond to numerical simulations. The dashed line corresponds to the theoretical prediction of the means of the cluster, as given by equation (16) of Result 3.1. The target Gaussian mixture $\rho_1$ is represented in red. The base zero-mean Gaussian density $\rho_0$ (dark blue) is split by the flow (7) into two clusters, which approach the target clusters (red) as time accrues . (**right**) $\sigma = 2$. PCA visualization of the generated density $\hat{\rho}_1$, by training the generative model on $n$ samples, for $n \in \{4, 8, 16, 32, 64\}$. Point clouds represent numerical simulations of the generative model. Crosses represent the theoretical predictions of Result 3.1 for the means of the clusters of $\hat{\rho}_1$, as given by equation (16) of Result 3.1 for $t = 1$. As the number of training samples $n$ increases, the generated clusters of $\hat{\rho}_1$ approach the target clusters of $\rho_1$, represented in red.

*with probability $1/2$, and to the opposite thereoef with probability $1/2$. In equation (20), the initial conditions are understood as $M_{t_0} = Q_{t_0}^\xi = 0$, and we have denoted $\delta t_k \equiv t_{k+1} - t_k$ for clarity. Furthermore, the orthogonal component $\boldsymbol{X}_{t_k}^\perp \in \text{span}(\boldsymbol{\mu}_{\text{emp.}}, \boldsymbol{\xi})^\perp$ obeys the simple linear recursion*

$$\boldsymbol{X}_{t_{k+1}}^\perp = \left[1 + \delta t_k \left(\dot{\beta}(t_k)\hat{c}_{t_k} + \frac{\dot{\alpha}(t_k)}{\alpha(t_k)}(1 - \hat{c}_{t_k}\beta(t_k))\right)\right] \boldsymbol{X}_{t_k}^\perp. \tag{21}$$

*Finally, the statistic $Q_{t_k} \equiv \|\boldsymbol{X}_{t_k}\|^2/d$ is given with high probability by*

$$Q_{t_k} = M_{t_k}^2(1 + \sigma^2/n) + n(Q_{t_k}^\xi)^2 + \prod_{\ell=0}^{k} \left[1 + \left(\dot{\beta}(t_\ell)\hat{c}_{t_\ell} + \frac{\dot{\alpha}(t_\ell)}{\alpha(t_\ell)}(1 - \hat{c}_{t_\ell}\beta(t_\ell))\right)\delta t_\ell\right]^2. \tag{22}$$

Equations (20),(21) and (22) of Remark 3.2 are consistent discretizations of the continuous flows (16),(17) and (18) of Result 3.1 respectively, and converge thereto in the limit of small discretization steps $\max_k \delta t_k \to 0$. A derivation of Remark 3.2 is detailed in Appendix B. An important consequence of Result 3.1 is that the transport of a sample $\boldsymbol{X}_0 \sim \rho_0$ by (7) factorizes into the low-dimensional deterministic evolution of its projection on the low-rank subspace $\text{span}(\boldsymbol{\mu}_{\text{emp.}}, \boldsymbol{\xi})$, as tracked by the two summary statistics $M_t, Q_t^\xi$, and the simple linear dynamics of its projection on the orthogonal space $\text{span}(\boldsymbol{\mu}_{\text{emp.}}, \boldsymbol{\xi})^\perp$. Result 3.1 thus reduces the high-dimensional flow (7) into a set of two scalar ordinary differential equations (16) and a simple homogeneous linear differential equation (17). The theoretical predictions of Result (3.1) and Remark 3.2 for the summary statistics $M_t, Q_t^\xi, Q_t$ are plotted in Fig. 2, and display convincing agreement with numerical simulations, corresponding to discretizing the flow (7) in $N = 100$ time steps, and training a separate network for each step as described in Section 1. A PCA visualization of the flow is further provided in Fig. 2 (middle).

Leveraging the simple characterization of Result 3.1, one is now in a position to characterize the generated distribution $\hat{\rho}_1$, which is the density effectively sampled by the generative model. In particular, Result 3.1 establishes that the distribution $\hat{\rho}_1$ is Gaussian over $\text{span}(\boldsymbol{\mu}_{\text{emp.}}, \boldsymbol{\xi})^\perp$ – since $\boldsymbol{X}_0^\perp$ is Gaussian and the flow is linear–, while the density in $\text{span}(\boldsymbol{\mu}_{\text{emp.}}, \boldsymbol{\xi})$ concentrates along the vector $\hat{\boldsymbol{\mu}}$ described by the components (16). The density $\hat{\rho}_1$ is thus described by a mixture of two

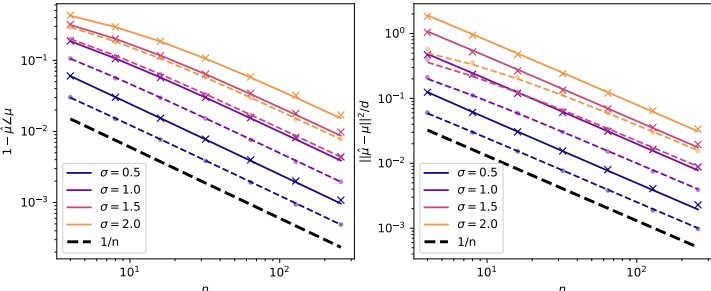

Figure 3: $\alpha(t) = 1 - t, \beta(t) = t, \varphi = \text{sign}$. Cosine asimilarity (left) and mean squared distance (right) between the mean $\hat{\boldsymbol{\mu}}$ of the generated mixture $\hat{\rho}_1$ and the mean $\boldsymbol{\mu}$ of the target density $\rho_1$, as a function of the number of training samples $n$, for various variances $\sigma$ of $\rho_1$. Solid lines represent the theoretical characterization of Corollary 3.3. Crosses represent numerical simulations of the generative model, by discretizing the differential equation (7) with step size $\delta t = 0.01$, and training a separate DAE for each time step using the `Pytorch` implementation of the full-batch Adam optimizer, with learning rate 0.04 and weight decay $\lambda = 0.1$ for 6000 epochs. All experiments were conducted in dimension $d = 5 \times 10^4$, and a single run is represented. Dashed lines indicate the performance of the Bayes-optimal estimator $\hat{\boldsymbol{\mu}}^\star$, as theoretically characterized in Remark 4.1. Dots indicate the performance of the PCA estimator, which is found as in Cui & Zdeborová (2023) to yield performances nearly identical to the Bayes-optimal estimator.

clusters, Gaussian along $d - 2$ directions, centered around $\pm\hat{\boldsymbol{\mu}}$. The following corollary provides a sharp characterization of the squared distance between the mean $\hat{\boldsymbol{\mu}}$ of the generated density $\hat{\rho}_1$ and the true mean $\boldsymbol{\mu}$ of the target density $\rho_1$.

**Corollary 3.3.** *(Mean squared error of the mean estimate) Let $\hat{\boldsymbol{\mu}}$ be the cluster mean of the density $\hat{\rho}_1$ generated by the (continuous) learnt flow (7). In the asymptotic limit described by Result 2.1, the squared distance between $\hat{\boldsymbol{\mu}}$ and the true mean $\boldsymbol{\mu}$ is given by*

$$\frac{1}{d}\|\hat{\boldsymbol{\mu}} - \boldsymbol{\mu}\|^2 = M_1^2 + n(Q_1^\xi)^2 + n\sigma^2(Q_1^\eta)^2 + 1 - 2M_1, \tag{23}$$

*with $M_1, Q_1^\xi, Q_1^\eta$ being the solutions of the ordinary differential equations (16) evaluated at time $t = 1$. Furthermore, the cosine similarity between $\hat{\boldsymbol{\mu}}$ and the true mean $\boldsymbol{\mu}$ is given by*

$$\hat{\boldsymbol{\mu}} \angle \boldsymbol{\mu} = \frac{M_1}{\sqrt{Q_1}}. \tag{24}$$

*Finally, both the Mean Squared Error (MSE) $^1/d\|\hat{\boldsymbol{\mu}} - \boldsymbol{\mu}\|^2$ (23) and the cosine asimilarity $1 - \hat{\boldsymbol{\mu}} \angle \boldsymbol{\mu}$ (24) decay as $\Theta_n(^1/n)$ for large number of samples $n$.*

The heuristic derivation of Corollary 3.3 is presented in Appendix A.1. The theoretical predictions of the learning metrics (23) and (24) are plotted in Fig. 3 as a function of the number of samples, along with the corresponding numerical simulations, and display a clear $\Theta_n(^1/n)$ decay, signalling the convergence of the generated density $\hat{\rho}_1$ to the true target density $\rho_1$ as the sample complexity accrues. A PCA visualization of this convergence is further presented in Fig.2 (right). Intuitively, this is because the DAE learns the empirical means up to a $\Theta_n(^1/n)$ component along $\boldsymbol{\xi}$, and that the empirical means itself converges to the true mean with rate $\Theta_n(^1/n)$. While we focus on the MSE for conciseness, the rate of convergence in terms of a variant of the squared gaussian mixture Wasserstein distance (Delon & Desolneux, 2020; Chen et al., 2018) can similarly be derived to be $\Theta_n(^1/n)$, see Appendix F.

## 4 BAYES-OPTIMAL BASELINE

Corollary 3.3 completes the study of the performance of the DAE-parametrized generative model. It is natural to wonder whether one can improve on the $\Theta_n(^1/n)$ rate that it achieves. A useful baseline to compare with is the Bayes-optimal estimator $\hat{\boldsymbol{\mu}}^\star$, yielded by Bayesian inference when in addition to the dataset $\mathcal{D} = \{\boldsymbol{x}_1^\mu\}_{\mu=1}^n$, the form of the distribution (8) and the variance $\sigma$ are known, but *not* the mean $\boldsymbol{\mu}$ —which for definiteness and without loss of generality will be assumed in this section to be have been drawn at random from $\mathcal{N}(0, \mathbb{I}_d)$. The following remark provides a tight characterization of the MSE achieved by this estimator.

**Remark 4.1.** *(**Bayes-optimal estimator of the cluster mean**) The Bayes-optimal estimator $\hat{\boldsymbol{\mu}}^\star$ of $\boldsymbol{\mu}$ assuming knowledge of the functional form of the target density (8), the cluster variance $\sigma$, and the training set $\mathcal{D}$, is defined as the minimizer of the average squared error*

$$\hat{\boldsymbol{\mu}}^\star = \underset{\nu}{\arg\inf} \mathbb{E}_{\boldsymbol{\mu}\sim\mathcal{N}(0,\mathbb{I}_d),\mathcal{D}\sim\rho_1^{\otimes n}} \|\nu(\mathcal{D}) - \boldsymbol{\mu}\|^2. \tag{25}$$

*In the asymptotic limit of Result 2.1, the Bayes-optimal estimator $\hat{\boldsymbol{\mu}}^\star(\mathcal{D})$ is parallel to the empirical mean $\boldsymbol{\mu}_{\mathrm{emp.}}$. Its component $m^\star \equiv \boldsymbol{\mu}_{\mathrm{emp.}}^\top \hat{\boldsymbol{\mu}}^\star(\mathcal{D})/d(1 + \sigma^2/n)$ concentrate asymptotically to*

$$m^\star = \frac{n}{n + \sigma^2}, \tag{26}$$

*Finally, with high probability, the Bayes-optimal MSE reads*

$$\frac{1}{d}\|\hat{\boldsymbol{\mu}}^\star(\mathcal{D}) - \boldsymbol{\mu}\|^2 = \frac{\sigma^2}{n + \sigma^2}. \tag{27}$$

*In particular, (27) implies that the optimal MSE decays as $\Theta_n(1/n)$.*

Remark 4.1, whose derivation is detailed in Appendix C, thus establishes that the Bayes-optimal MSE decays as $\Theta_n(1/n)$ with the number of available training samples. Note that while the Bayes-optimal estimator is colinear to the empirical mean, it is differs therefrom by a non-trivial multiplicative factor. On the other hand, the $\Theta_n(1/n)$ rate is intuitively due to the $\Theta_n(1/n)$ convergence of the empirical mean to the true mean. Contrasting to Corollary 3.3 for the MSE associated to the mean $\hat{\boldsymbol{\mu}}$ of the density $\hat{\rho}_1$ learnt by the generative model, it follows that *the latter achieves the Bayes-optimal learning rate*. The Bayes-optimal MSE (27) predicted by Remark 4.1 is plotted in dashed lines in Fig. 3, alongside the MSE achieved by the generative model (see Corollary 3.3). The common $1/n$ decay rate is also plotted in dashed black for comparison. Finally, we observe that the estimate of $\boldsymbol{\mu}$ inferred by PCA, plotted as dots in Fig. 3, leads to a cosine similarity which is very close to the Bayes-optimal one, echoing the findings of Cui & Zdeborová (2023) in another asymptotic limit. We however stress an important distinction between the generative model analyzed in previous sections and the Bayes and PCA estimators dicussed in the present section. The generative model is tasked with estimating the full distribution $\rho_1$ only from data, while being completely agnostic thereof. In contrast, PCA and Bayesian inference only offer an estimate of the cluster mean, and require an exact oracle knowledge of its functional form (8) and the cluster variance $\sigma$. They do *not*, therefore, constitute generative models and are only discussed in the present section as insightful baselines.

It is a rather striking finding that the DAE (9) succeeds in approximately sampling from $\rho_1$(8) when trained on but $n = \Theta_d(1)$ samples –instead of simply generating back memorized training samples–, and further displays information-theoretically optimal learning rates. The answer to this puzzle lies in the fact that the architecture (9) is very close to the functional form of the exact velocity field $b$ (1), as further detailed in Appendix B (see equation (67)), and is therefore implicitly biased towards learning the latter – while also not being expressive enough to too detrimentally overfit. A thorough exploration of this form of inductive bias for more complex architectures is an important and fascinating entreprise, which falls out of the scope of the present manuscript and is left for future work.

## CONCLUSION

We conduct a tight end-to-end asymptotic analysis of estimating and sampling a binary Gaussian mixture using a flow-based generative model, when the flow is parametrized by a shallow auto-encoder. We provide sharp closed-form characterizations for the trained weights of the network, the learnt velocity field, a number of summary statistics tracking the generative flow, and the distance between the mean of the generated mixture and the mean of the target mixture. The latter is found to display a $\Theta_n(1/n)$ decay rate, where $n$ is the number of samples, which is further shown to be the Bayes-optimal rate. In contrast to most studies of flow-based generative models in high dimensions, the learning and sampling processes are jointly and sharply analyzed in the present manuscript, which affords the possibility to explicitly investigate the effect of a limited sample complexity at the level of the generated density.

ACKNOWLEDGEMENT

We thank Michael Albergo, Nicholas Boffi, Joan Bruna, Arthur Jacot and Ahmed El Alaoui for insightful discussions. Part of this work was done during HC's visit in the Courant Institute in March

2023. We acknowledge funding from the Swiss National Science Foundation grants OperaGOST (grant number 200390) and SMArtNet (grant number 212049). EVE is supported by the National Science Foundation under awards DMR-1420073, DMS-2012510, and DMS-2134216, by the Simons Collaboration on Wave Turbulence, Grant No. 617006, and by a Vannevar Bush Faculty Fellowship.

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

# A   DERIVATION OF RESULT 2.1

In this appendix, we detail the heuristic derivation of Result (2.1), which provides a sharp asymptotic characterization of the parameters $\hat{c}_t, \hat{w}_t$ of the DAE (9) minimizing the time $t$ risk $\hat{\mathcal{R}}_t$ (10). In the following, the time index $t \in [0,1]$ is considered fixed.

In order to characterize observables depending on the minimizers $\hat{c}_t, \hat{w}_t$ of the risk $\hat{\mathcal{R}}_t$ (10), observe that for any test function $\phi(c_t, w_t)$:

$$\phi(\hat{c}_t, \hat{w}_t) = \lim_{\gamma \to \infty} \frac{1}{Z} \int dw dc \phi(c,w) e^{-\gamma \hat{\mathcal{R}}_t(w,c)}, \tag{28}$$

where the normalization $Z$ is

$$\mathcal{Z}(\mathcal{D}) = \int dc \, dw e^{-\frac{\gamma}{2} \left\| x_1^\mu - \left[ c \times (\beta(t) x_1^\mu + \alpha(t) x_0^\mu) + w\varphi(w^\top(\beta(t) x_1^\mu + \alpha(t) x_0^\mu)) \right] \right\|^2 - \frac{\gamma\lambda}{2} \|w\|^2}. \tag{29}$$

We emphasized the dependence on the train set $\mathcal{D} = \{x_0^\mu, x_t^\mu\}_{\mu=1}^n$. $\ln Z(\mathcal{D})$ can then be studied as a moment generating function, and integrals of the form (28) deduced therefrom. In the following, we therefore seek to establish an asymptotic characterization of $\ln Z(\mathcal{D})$.

An important observation lies in the fact that the argument $w^\top x$ of the activation $\varphi$ of the DAE is expected in high dimensions $d \to \infty$ to be very large. In particular, we shall self-consistently establish that it is more precisely scaling like $\Theta_d(d)$. As a result, only the asymptotic behaviour in $\pm\infty$ of $\varphi$ matters, and by assumption $\varphi(w^\top x) \approx \mathrm{sign}(w^\top x)$ asymptotically. We shall therefore self-consistently take $\varphi = \mathrm{sign}$ in the following.

## A.1   COMPUTATION OF THE PARTITION FUNCTION

In the following, for clarity, we use the decomposition $x_1^\mu = s^\mu \mu + z^\mu$, introduced below (8) in the main text, with $s^\mu \in \{-1, +1\}$ and $z^\mu \sim \mathcal{N}(0, \sigma^2 \mathbb{I}_d)$. Under these notations, the partition function reads:

$$\mathcal{Z}(\mathcal{D}) = \int dc \, dw e^{-\frac{\gamma d}{2} \sum\limits_{\mu=1}^{n} \frac{\|w\|^2}{d} \mathrm{sign}\left( w^\top(\beta(t)(s^\mu \mu + z^\mu) + \alpha(t) x_0^\mu) \right)^2}$$

$$\times e^{\gamma d \sum\limits_{\mu=1}^{n} \mathrm{sign}\left( w^\top(\beta(t)(s^\mu \mu + z^\mu) + \alpha(t) x_0^\mu) \right) \frac{w^\top((1-c\beta(t))(s^\mu \mu + z^\mu) - c\alpha(t) x_0^\mu)}{d}} \times e^{-\frac{\gamma\lambda}{2} \|w\|^2}$$

$$\times e^{-\frac{\gamma d}{2} \sum\limits_{\mu=1}^{n} \left[ (1-\beta(t)c)^2 \frac{\|\mu^\mu\|^2 + \|z^\mu\|^2 + 2s^\mu \mu^\top z^\mu}{d} + c^2 \frac{\|\alpha(t) x_0^\mu\|^2}{d} + 2b(\beta(t)c-1) \frac{s^\mu \mu^\top \alpha(t) x_0^\mu + \eta^{\mu\top} \alpha(t) x_0^\mu}{d} \right]} \tag{30}$$

Note that we benignly introduced a $1/d$ factor inside the sign function $\mathrm{sign}$. One is now in position to introduce the overlaps

$$q \equiv \frac{\|w\|^2}{2}, \qquad q_\xi^\mu \equiv s^\mu \frac{w^\top x_0^\mu}{d}, \qquad q_\eta^\mu \equiv s^\mu \frac{w^\top z^\mu}{d}, \qquad m \equiv \frac{w^\top \mu}{d}. \tag{31}$$

Note that because of $n = \Theta(1)$ there is a finite number of these such overlaps. Besides, note that our starting assumption that the argument $w^\top x$ of the activation $\varphi$ is $\Theta_d(d)$ translates into the fact that all these order parameters should be $\Theta_d(1)$, which we shall self-consistently show to be indeed the case. The partition function then reads

$$\mathcal{Z}(\mathcal{D}) = \int dc \, dm d\hat{m} \, dq d\hat{q} \prod_{\mu=1}^{n} d\hat{q}_\eta^\mu d\hat{q}_\eta^\mu \, dq_\xi^\mu e^{\frac{d}{2} \hat{q} q + d\hat{m} m + d \sum\limits_{\mu=1}^{n} (\hat{q}_\xi^\mu q_\xi^\mu + \hat{q}_\eta^\mu q_\eta^\mu)}$$

$$\int dw e^{-\frac{\gamma\lambda}{2} \|w\|^2 - \frac{\hat{q}}{2} \|w\|^2 - \left( \hat{m}\mu + \sum\limits_{\mu=1}^{n} (\hat{q}_\xi^\mu s^\mu x_0^\mu + \hat{q}_\eta^\mu s^\mu z^\mu) \right)^\top w} e^{-\frac{\gamma d}{2} \sum\limits_{\mu=1}^{n} \left[ (1+\sigma^2)(1-\beta(t)c)^2 + c^2\alpha(t)^2 \right]}$$

$$e^{-\frac{\gamma d}{2} \sum\limits_{\mu=1}^{n} \left[ q \, \mathrm{sign}\left( \beta(t)(m+q_\eta^\mu) + \alpha(t) q_\xi^\mu \right)^2 - 2 \, \mathrm{sign}\left( \beta(t)(m+q_\eta^\mu) + \alpha(t) q_\xi^\mu \right) \left[ (1-c\beta(t))(m+q_\eta^\mu) - c\alpha(t) q_\xi^\mu \right] \right]}. \tag{32}$$

Therefore

$$
\mathcal{Z}(\mathcal{D}) = \int dc\, dm d\hat{m}\, dq d\hat{q} \prod_{\mu=1}^{n} d\hat{q}_\xi^\mu dq_\eta^\mu d\hat{q}_\eta^\mu\, dq_\xi^\mu
$$

$$
e^{\frac{d}{2}\hat{q}q + d\hat{m}m + d\sum_{\mu=1}^{n}(\hat{q}_\xi^\mu q_\xi^\mu + \hat{q}_\eta^\mu q_\eta^\mu) + \frac{d}{2(\gamma\lambda+\hat{q})}\frac{1}{d}\left\|\hat{m}\mu + \sum_{\mu=1}^{n}(\hat{q}_\xi^\mu s^\mu x_0^\mu + \hat{q}_\eta^\mu s^\mu z^\mu)\right\|^2}
$$

$$
e^{-\frac{\gamma d}{2}\sum_{\mu=1}^{n}\left[q\,\mathrm{sign}\left(\beta(t)(m+q_\eta^\mu) + \alpha(t)q_\xi^\mu\right)^2 - 2\,\mathrm{sign}\left(\beta(t)(m+q_\eta^\mu) + \alpha(t)q_\xi^\mu\right)\left[(1-c\beta(t))(m+q_\eta^\mu) - c\alpha(t)q_\xi^\mu\right]\right]}
$$

$$
e^{-\frac{\gamma d}{2}\sum_{\mu=1}^{n}\left[(1+\sigma^2)(1-\beta(t)c)^2 + c^2\alpha(t)^2\right] - \frac{d}{2}\ln(\gamma\lambda+\hat{q})}. \tag{33}
$$

The last term in the first exponent can be further simplified as

$$
\frac{1}{d}\left\|\hat{m}\mu + \sum_{\mu=1}^{n}(\hat{q}_\xi^\mu s^\mu x_0^\mu + \hat{q}_\eta^\mu s^\mu z^\mu)\right\|^2 = \hat{m}^2 + \sum_{\mu=1}^{n}\left[(\hat{q}_\xi^\mu)^2 + (\hat{q}_\eta^\mu)^2\sigma^2\right]
$$

$$
+ 2\sum_{\mu=1}^{n}\left[\hat{q}_\xi^\mu s^\mu \frac{\mu^\top x_0^\mu}{d} + \hat{q}_\eta^\mu s^\mu \frac{\mu^\top z^\mu}{d}\right]
$$

$$
+ \sum_{\mu,\nu=1}^{n} s^\mu s^\nu \left[\hat{q}_\xi^\mu \hat{q}_\xi^\nu \frac{(x_0^\nu)^\top x_0^\mu}{d} + \hat{q}_\eta^\mu \hat{q}_\eta^\nu \frac{(z^\nu)^\top z^\mu}{d} + \hat{q}_\xi^\mu \hat{q}_\eta^\nu \frac{(z^\nu)^\top x_0^\mu}{d}\right]
$$

$$
= \hat{m}^2 + \sum_{\mu=1}^{n}\left[(\hat{q}_\xi^\mu)^2 + (\hat{q}_\eta^\mu)^2\sigma^2\right] + \mathcal{O}\left(1/\sqrt{d}\right), \tag{34}
$$

with the last line holding with high probability, using the fact that since $z, x_0$ are two independently drawn standard Gaussian vectors $z^\top x_0/d = \Theta_d(1/\sqrt{d})$ with high probability. Finally,

$$
\mathcal{Z}(\mathcal{D}) = \int dc\, dm d\hat{m}\, dq d\hat{q} \prod_{\mu=1}^{n} d\hat{q}_\xi^\mu dq_\eta^\mu d\hat{q}_\eta^\mu\, dq_\xi^\mu
$$

$$
e^{\frac{d}{2}\hat{q}q + d\hat{m}m + d\sum_{\mu=1}^{n}(\hat{q}_\xi^\mu q_\xi^\mu + \hat{q}_\eta^\mu q_\eta^\mu) + \frac{d}{2(\gamma\lambda+\hat{q})}\left[\hat{m}^2 + \sum_{\mu=1}^{n}\left[(\hat{q}_\xi^\mu)^2 + (\hat{q}_\eta^\mu)^2\sigma^2\right]\right]}
$$

$$
e^{-\frac{\gamma d}{2}\sum_{\mu=1}^{n}\left[q\,\mathrm{sign}\left(\beta(t)(m+q_\eta^\mu) + \alpha(t)q_\xi^\mu\right)^2 - 2\,\mathrm{sign}\left(\beta(t)(m+q_\eta^\mu) + \alpha(t)q_\xi^\mu\right)\left[(1-c\beta(t))(m+q_\eta^\mu) - c\alpha(t)q_\xi^\mu\right]\right]}
$$

$$
e^{-\frac{\gamma d}{2}\sum_{\mu=1}^{n}\left[(1+\sigma^2)(1-\beta(t)c)^2 + c^2\alpha(t)^2\right] - \frac{d}{2}\ln(\gamma\lambda+\hat{q})} \tag{35}
$$

Since all the terms in the exponent of the integrand scale like $d$, in the asymptotic limit $d \to \infty$ the integral can be computed using a Laplace approximation.

## A.2 SAMPLE-SYMMETRIC ANSATZ

The partition function is given by taking the saddle point in (35). This involves a maximization problem over $4n + 5$ variables. Note that since $n = \Theta_d(1)$, this is a low dimensional – thus a priori tractable – optimization problem, but which nevertheless remains cumbersome. However, the symmetries of the problem make it possible to determine the form of the maximizer, and thus drastically simplify the optimization problem. Note that indeed, asymptotically, the vectors $\mu, \{x_0^\mu, z^\mu\}_{\mu=1}^{n}$ involved in the definition of the overlaps $m, \{q_\xi^\mu, q_\eta^\mu\}_{\mu=1}^{n}$ (31) are all mutually asymptotically orthogonal – i.e. they have vanishing cosine similarity. Therefore, the parameters $m, \{q_\xi^\mu, q_\eta^\mu\}_{\mu=1}^{n}$ can be considered as independent variables. Since furthermore all the samples play interchangeable roles in high dimensions – in that all data points are asymptotically at the same angle with the cluster mean $\mu$, which is the only relevant direction of the problem–, one can look for the saddle point assuming the symmetric ansatz

$$
\forall\mu,\ q_\xi^\mu = q_\xi,\ \hat{q}_\xi^\mu = \hat{q}_\xi, \tag{36}
$$

$$
\forall\mu,\ q_\eta^\mu = q_\eta,\ \hat{q}_\eta^\mu = \hat{q}_\eta. \tag{37}
$$

This ansatz is further validated in numerical experiments, when training a DAE with the `Pytorch` implementation of the Adam optimizer. Under this ansatz, the partition function reduces to

$$
\mathcal{Z}(\mathcal{D}) = \int dc\, dm\, d\hat{m}\, dq\, d\hat{q}\, d\hat{q}_\xi\, dq_\eta\, d\hat{q}_\eta\, dq_\xi\, e^{\frac{d}{2}\hat{q}q + d\hat{m}m + dn(\hat{q}_\xi q_\xi + \hat{q}_\eta q_\eta) + \frac{d}{2(\gamma\lambda+\hat{q})}\left[\hat{m}^2 + n(\hat{q}_\xi^2 + \hat{q}_\eta^2\sigma^2)\right]}
$$

$$
e^{-\frac{\gamma d}{2}n\left[q\,\mathrm{sign}\left(\beta(t)(m+q_\eta)+\alpha(t)q_\xi\right)^2 - 2\,\mathrm{sign}\left(\beta(t)(m+q_\eta)+\alpha(t)q_\xi\right)\left[(1-c\beta(t))(m+q_\eta)-c\alpha(t)q_\xi\right]+(1+\sigma^2)(1-\beta(t)c)^2+c^2\alpha(t)^2\right]}
$$

$$
e^{-\frac{d}{2}\ln(\gamma\lambda+\hat{q})} \tag{38}
$$

Note that the exponent is now *independent of the dataset* $\mathcal{D}$. In other words, in the regime $d \to \infty$, $n = \Theta_d(1)$, the log partition function concentrates with respect to the randomness associated with the sampling of the training set. The effective action (log partition function) therefore reads

$$
\ln \mathcal{Z}(\mathcal{D}) = \operatorname*{extr}_{c,\hat{q},q,\hat{m},m,\hat{q}_{\eta,\xi},q_{\eta,\xi}} \frac{1}{2}\hat{q}q + \hat{m}m + n(\hat{q}_\xi q_\xi + \hat{q}_\eta q_\eta) + \frac{1}{2(\gamma\lambda+\hat{q})}\left[\hat{m}^2 + n(\hat{q}_\xi^2 + \hat{q}_\eta^2\sigma^2)\right]
$$

$$
- \frac{\alpha}{2}n\left[q\,\mathrm{sign}(\beta(t)(m+q_\eta)+\alpha(t)q_\xi)^2 - 2\,\mathrm{sign}(\beta(t)(m+q_\eta)+\alpha(t)q_\xi)[(1-c\beta(t))(m+q_\eta)-c\alpha(t)q_\xi]\right]
$$

$$
- \frac{\gamma n}{2}\left[(1+\sigma^2)(1-\beta(t)c)^2 + c^2\alpha(t)^2\right] - \frac{1}{2}\ln(\gamma\lambda+\hat{q}) \tag{39}
$$

This expression has to be extremized with respect to $c, \hat{q}, q, \hat{m}, m, \hat{q}_{\eta,\xi}, q_{\eta,\xi}$ in the $\gamma \to \infty$ limit. Rescaling the conjugate variables as

$$
\gamma\hat{q} \leftarrow \hat{q}, \qquad\qquad \gamma\hat{q}_{\eta,\xi} \leftarrow \hat{q}_{\eta,\xi}, \qquad\qquad \gamma\hat{m} \leftarrow \hat{m} \tag{40}
$$

the action becomes, in the $\gamma \to \infty$ limit (changing for readability the conjugates $\hat{m}, \hat{q}_{\eta,\xi} \to -\hat{m}, -\hat{q}_{\eta,\xi}$):

$$
\ln \mathcal{Z}(\mathcal{D}) = \operatorname*{extr}_{c,\hat{q},q,\hat{m},m,\hat{q}_{\eta,\xi},q_{\eta,\xi}} \frac{1}{2}\hat{q}q - \hat{m}m - n(\hat{q}_\xi q_\xi + \hat{q}_\eta q_\eta) + \frac{1}{2(\lambda+\hat{q})}\left[\hat{m}^2 + n(\hat{q}_\xi^2 + \hat{q}_\eta^2\sigma^2)\right]
$$

$$
- \frac{n}{2}\left[q\,\mathrm{sign}(\beta(t)(m+q_\eta)+\alpha(t)q_\xi)^2 - 2\,\mathrm{sign}(\beta(t)(m+q_\eta)+\alpha(t)q_\xi)[(1-c\beta(t))(m+q_\eta)-c\alpha(t)q_\xi]\right]
$$

$$
- \frac{n}{2}\left[(1+\sigma^2)(1-\beta(t)c)^2 + c^2\alpha(t)^2\right] \tag{41}
$$

## A.3 SADDLE-POINT EQUATIONS

The extremization of $\ln \mathcal{Z}(\mathcal{D})$ can be alternatively written as zero-gradient equations on each of the parameters the extremization is carried over, yielding

$$
\begin{cases}
q = \frac{\hat{m}^2 + n(\hat{q}_\xi^2 + \hat{q}_\eta^2\sigma^2)}{(\lambda+\hat{q})^2} \\
m = \frac{\hat{m}}{\lambda+\hat{q}} \\
q_\xi = \frac{\hat{q}_\xi}{\lambda+\hat{q}} \\
q_\eta = \frac{\hat{q}_\eta\sigma^2}{\lambda+\hat{q}}
\end{cases}
\qquad
\begin{cases}
\nu \equiv \beta(t)(m+q_\eta) + \alpha(t)q_\xi \\
\hat{q} = n \\
\hat{m} = n\,\mathrm{sign}(\nu)(1 - c\beta(t)) \\
\hat{q}_\eta = \frac{\hat{m}}{n} \\
\hat{q}_\xi = -c\alpha(t)\,\mathrm{sign}(\nu) \\
c = \frac{(1+\sigma^2)\beta(t) - \mathrm{sign}(\nu)(\beta(t)(m+q_\eta)+\alpha(t)q_\xi)}{\alpha(t)^2 + \beta(t)^2(1+\sigma^2)}
\end{cases}
\tag{42}
$$

Note the identity

$$
q = m^2 + n(q_\xi^2 + q_\eta^2/\sigma^2) \tag{43}
$$

which follows from the asymptotic orthogonality of the vectors and the Pythagorean theorem. This implies in particular that the square norm of $\hat{w}$, as measure by $q$, is only the sum of its projections along $\mu$ (corresponding to $m$) $\xi$ (corresponding to $q_\xi$) and $\eta$ ($q_\eta$). Therefore, the norm of the orthogonal projection of $\hat{w}$ with respect to $\mathrm{span}(\mu, \eta, \xi)$ is asymptotically vanishing. In other words, $\hat{w}$ is asymptotically contained in $\mathrm{span}(\mu, \eta, \xi)$.

Remark that the symmetry $m, \hat{m}, q_{\eta,\xi}, \hat{q}_{\eta,\xi} \to -m, -\hat{m}, -q_{\eta,\xi}, -\hat{q}_{\eta,\xi}$ leaves equations (42) unchanged, meaning that if $m, \hat{m}, q_{\eta,\xi}, \hat{q}_{\eta,\xi}$ is a solution to the saddle-point equations, so is $m, \hat{m}, q_{\eta,\xi}, \hat{q}_{\eta,\xi}$. This is due to the symmetry between the clusters in the target density (8), as

$\mu \to -\mu$ yields the same model. As a convention, we can thus suppose without loss of generality $\nu \geq 0$ in (42). (42) then simplifies to

$$
\begin{cases}
q = \frac{\hat{m}^2 + n(\hat{q}_\xi^2 + \hat{q}_\eta^2 \sigma^2)}{(\lambda + \hat{q})^2} \\
m = \frac{\hat{m}}{\lambda + \hat{q}} \\
q_\xi = \frac{\hat{q}_\xi}{\lambda + \hat{q}} \\
q_\eta = \frac{\hat{q}_\eta \sigma^2}{\lambda + \hat{q}}
\end{cases}
\qquad
\begin{cases}
\hat{q} = n \\
\hat{m} = n(1 - c\beta(t)) \\
\hat{q}_\eta = \frac{\hat{m}}{n} \\
\hat{q}_\xi = -\alpha(t)c \\
c = \frac{(1 + \sigma^2)\beta(t) - (\beta(t)(m + q_\eta) + \alpha(t)q_\xi)}{\alpha(t)^2 + \beta(t)^2(1 + \sigma^2)}
\end{cases}.
\tag{44}
$$

The skip connection strength $c$ thus satisfies the self-consistent equation

$$
c = \frac{(1 + \sigma^2)\beta(t) - \frac{1}{\lambda + n}(\beta(t)(1 - \beta(t)c)(n + \sigma^2) - \alpha(t)^2 c)}{\alpha(t)^2 + \beta(t)^2(1 + \sigma^2)},
\tag{45}
$$

which can be solved as

$$
c = \frac{\beta(t)(\lambda(1 + \sigma^2) + (n - 1)\sigma^2)}{\alpha(t)^2(\lambda + n - 1) + \beta(t)^2(\lambda(1 + \sigma^2) + (n - 1)\sigma^2)}
\tag{46}
$$

which recovers equation (11) of Result 2.1. Plugging this expression back to (44), and redefining $\sigma^2 q_\eta \leftarrow q_\eta$, yields

$$
\begin{cases}
m_t = \frac{n}{\lambda + n} \frac{\alpha(t)^2(\lambda + n - 1)}{\alpha(t)^2(\lambda + n - 1) + \beta(t)^2(\lambda(1 + \sigma^2) + (n - 1)\sigma^2)} \\
q_t^\eta = \frac{\sigma^2}{\lambda + n} \frac{\alpha(t)^2(\lambda + n - 1)}{\alpha(t)^2(\lambda + n - 1) + \beta(t)^2(\lambda(1 + \sigma^2) + (n - 1)\sigma^2)} \\
q_t^\xi = -\frac{1}{\lambda + n} \frac{\alpha(t)\beta(t)(\lambda(1 + \sigma^2) + (n - 1)\sigma^2)}{\alpha(t)^2(\lambda + n - 1) + \beta(t)^2(\lambda(1 + \sigma^2) + (n - 1)\sigma^2)}
\end{cases}
\tag{47}
$$

We have added subscripts $t$ to emphasize the dependence on the time index $t$. Note that for $t > 0$, $\nu > 0$, which is self-consistent. For $t = 0$, $\nu = 0$ and the sign function in equation (42) becomes ill-defined, signalling that the extremum of equation (41) ceases to be a critical point (i.e. differentiable). However, one expects the extremum to still be given by the $t = 0$ limit of equation (42), as there is a priori no singularity in the learning problem for $t = 0$. This remark, together with (47), recovers equation (14) from Result 2.1. $\qquad\square$

## A.4 METRICS

Result 2.1 provides a tight characterization of the skip conneciton strength $\hat{c}_t$ and of the vector $\hat{w}_t$. The performance of the trained DAE $f_{\hat{c}_t, \hat{w}_t}$ (9) as a denoiser can be further quantified with a number of metrics, for which we also provide sharp asymptotic characterizations below, for completeness.

**Result A.1.** *(MSE) The test MSE of the learnt denoiser $f_{\hat{c}_t, \hat{w}_t}$ is defined as the test error associated to the risk $\hat{R}_t$ (10)*

$$
\mathrm{mse}_t \equiv \mathbb{E}_{x_1 \sim \rho_1, x_0 \sim \rho_0} \left\| f_{\hat{c}_t, \hat{w}_t}(\alpha(t)x_0 + \beta(t)x_1) - x_1 \right\|^2.
\tag{48}
$$

*In the same asymptotic limit as Result 2.1 in the main text, this metric is sharply characterized by the closed-form formula*

$$
\mathrm{mse}_t = m_t^2 + n((q_t^\xi)^2 + (q_t^\eta)^2 \sigma^2) - 2(1 - \hat{c}_t\beta(t))m_t + (1 - \hat{c}_t\beta(t))^2(1 + \sigma^2) + \hat{c}_t^2 \alpha(t)^2
\tag{49}
$$

*where $\hat{c}_t, m_t, q_t^\xi, q_t^\eta$ were defined in Result 2.1. Furthermore, the MSE (48) is lower-bounded by the oracle MSE*

$$
\mathrm{mse}_t^\star \equiv \mathbb{E}_{x_1 \sim \rho_1, x_0 \sim \rho_0} \left\| f_t^\star(\alpha(t)x_0 + \beta(t)x_1) - x_1 \right\|^2,
\tag{50}
$$

*where the oracle denoiser follows from an application of Tweedie's formula Efron (2011); Albergo et al. (2023) as*

$$
f_t^\star(x) = \frac{\beta(t)\sigma^2}{\alpha(t)^2 + \beta(t)^2\sigma^2}x + \frac{\alpha(t)^2}{\alpha(t)^2 + \beta(t)^2\sigma^2}\mu \times \tanh\left(\frac{\beta(t)}{\alpha(t)^2 + \beta(t)^2\sigma^2}\mu^\top x\right).
\tag{51}
$$

*Finally, the oracle MSE $\mathrm{mse}_t^\star$ admits the following asymptotic characterization:*

$$
\mathrm{mse}_t^\star = \alpha(t)^4 \sigma^2 \frac{\alpha(t)^2 + \sigma^2(1 - \alpha(t)^2)}{(\sigma^2\beta(t)^2 + \alpha(t)^2)^2}
\tag{52}
$$

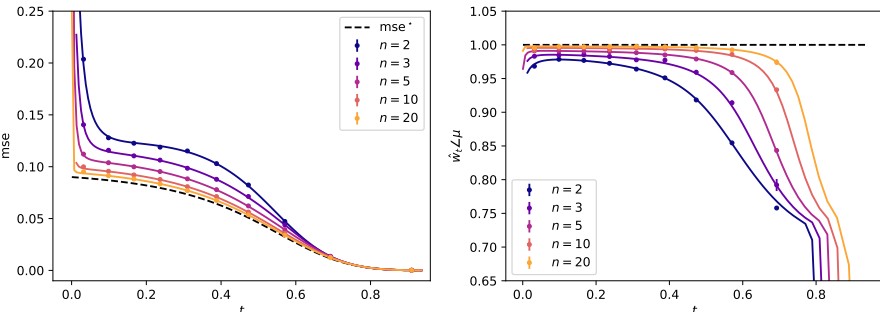

Figure 4: $\sigma = 0.3, \lambda = 0.1, \alpha(t) = \cos(\pi t/2), \beta(t) = \sin(\pi t/2)$. Solid lines: theoretical predictions for the MSE of Result A.1 (left) and the cosine similarity of Result A.2 (right). Different colors correspond to different number of samples $n$. Dots: numerical simulations, corresponding to training the DAE (9) on the risk (10) using the `Pytorch` implementation of full-batch Adam, with learning rate 0.01 over 2000 epochs and weight decay $\lambda = 0.1$. The experimental points correspond to a single instance of the model, and were collected in dimension $d = 500$. In the left plot, the dashed line represent the oracle baseline (52).

*Derivation of Result A.1* We begin by detailing the characterizaiton of the DAE MSE (49):

$$\mathrm{mse}_t = \frac{1}{d}\mathbb{E}_{x_1,x_0}\langle\left\|x_1 - \left[\hat{c}_t \times (\beta(t)x_1 + \alpha(t)x_0) + \hat{w}_t\,\mathrm{sign}\left(\hat{w}_t^\top(\beta(t)x_1 + \alpha(t)x_0)\right)\right]\right\|^2$$

$$= m_t^2 + n((q_t^\xi)^2 + (q_t^\eta)^2\sigma^2) - 2\,\mathrm{sign}(\beta(t)m_t)(1 - \hat{c}_t\beta(t))m_t + (1 - \hat{c}_t\beta(t))^2(1 + \sigma^2) + \hat{c}_t^2\alpha(t)^2$$

$$= m_t^2 + n((q_t^\xi)^2 + (q_t^\eta)^2\sigma^2) - 2(1 - \hat{c}_t\beta(t))m_t + (1 - \hat{c}_t\beta(t))^2(1 + \sigma^2) + \hat{c}_t^2\alpha(t)^2 \qquad (53)$$

which recovers (49) of Result A.1. (51) follows directly from an application of Tweedie's formula Efron (2011). The associated MSE (52) can be derived as

$$\mathrm{mse}^\star = \frac{\alpha(t)^4(1 + \sigma^2) + \sigma^4\alpha(t)^2(1 - \alpha(t)^2)}{(\sigma^2\beta(t)^2 + \alpha(t)^2)^2} + \frac{\alpha(t)^4}{(\sigma^2\beta(t)^2 + \alpha(t)^2)^2}\left[\mathrm{sign}\left(\frac{\beta(t)^2}{\sigma^2\beta(t)^2 + \alpha(t)^2}\right)^2\right]$$

$$- \frac{2\alpha(t)^2}{\sigma^2\beta(t)^2 + \alpha(t)^2}\left[\mathrm{sign}\left(\frac{\beta(t)^2}{\sigma^2\beta(t)^2 + \alpha(t)^2}\right)\right] \times \frac{\alpha(t)^2}{\sigma^2 + \alpha(t)^2 - \sigma^2\alpha(t)^2}$$

$$= \frac{\alpha(t)^4(1 + \sigma^2) + \sigma^4\alpha(t)^2(1 - \alpha(t)^2)}{(\sigma^2\beta(t)^2 + \alpha(t)^2)^2} - \frac{\alpha(t)^4}{(\sigma^2\beta(t)^2 + \alpha(t)^2)^2} = \frac{\alpha(t)^4\sigma^2 + \sigma^4\alpha(t)^2(1 - \alpha(t)^2)}{(\sigma^2\beta(t)^2 + \alpha(t)^2)^2}$$

$$(54)$$

which concludes the derivation of Result A.1 $\qquad\qquad\square$

**Result A.2.** *The cosine similarity $\hat{w}_t\angle\mu \equiv \hat{w}_t^\top\mu/\|\hat{w}_t\|\|\mu\|$ admits the asymptotic characterization*

$$\hat{w}_t\angle\mu = \frac{m_t}{\sqrt{m_t^2 + n((q_t^\xi)^2 + (q_t^\eta)^2\sigma^2}} \qquad (55)$$

*where $m_t, q_t^\xi, q_t^\eta$ are characterized in Result 2.1.*

Result A.2 follows directly from the definition of the summary statistics (13).

These metrics are plotted in Fig. 4 and contrasted to numerical simulations, corresponding to training the network (9) using the `Pytorch` implementation of full-batch Adam.

## B    DERIVATION OF RESULT 3.1

In this Appendix, we detail the heuristic derivation of Result 3.1. Given an initial condition $X_0 \sim \rho_0$, a sample follows the transport (7)

$$\frac{d}{dt}X_t = \left(\dot{\beta}(t)\hat{c}_t + \frac{\dot{\alpha}(t)}{\alpha(t)}(1 - \hat{c}_t\beta(t))\right)X_t + \left(\dot{\beta}(t) - \frac{\dot{\alpha}(t)}{\alpha(t)}\beta(t)\right)\hat{w}_t\,\mathrm{sign}(\hat{w}_t^\top X_t) \qquad (56)$$

driven by the learnt velocity field $\hat{b}$ (6). This follows from Result 2.1 and (7). Taking scalar products with $\mu, \xi, \eta$,

$$
\begin{cases}
\frac{d}{dt}\frac{X_t^\top \mu}{d} = \left(\dot{\beta}(t)\hat{c}_t + \frac{\dot{\alpha}(t)}{\alpha(t)}(1-\hat{c}_t\beta(t))\right)\frac{X_t^\top \mu}{d} + \left(\dot{\beta}(t)-\frac{\dot{\alpha}(t)}{\alpha(t)}\beta(t)\right)\text{sign}(\hat{w}_t^\top X_t)\frac{\hat{w}_t^\top \mu}{d} \\
\frac{d}{dt}\frac{X_t^\top \xi}{nd} = \left(\dot{\beta}(t)\hat{c}_t + \frac{\dot{\alpha}(t)}{\alpha(t)}(1-\hat{c}_t\beta(t))\right)\frac{X_t^\top \xi}{nd} + \left(\dot{\beta}(t)-\frac{\dot{\alpha}(t)}{\alpha(t)}\beta(t)\right)\text{sign}(\hat{w}_t^\top X_t)\frac{\hat{w}_t^\top \xi}{nd} \\
\frac{d}{dt}\frac{X_t^\top \eta}{nd\sigma^2} = \left(\dot{\beta}(t)\hat{c}_t + \frac{\dot{\alpha}(t)}{\alpha(t)}(1-\hat{c}_t\beta(t))\right)\frac{X_t^\top \eta}{nd\sigma^2} + \left(\dot{\beta}(t)-\frac{\dot{\alpha}(t)}{\alpha(t)}\beta(t)\right)\text{sign}(\hat{w}_t^\top X_t)\frac{\hat{w}_t^\top \eta}{nd\sigma^2}
\end{cases}. \quad (57)
$$

It is reasonable to assume the sign $\text{sign}(\hat{w}_t^\top X_t)$ stays constant during the transport, and therefore takes value $\pm 1$ with equal probability $1/2$, according to the initial condition $X_0$. This is an heuristic assumption which is further confirmed numerically. Finally, plugging the definitions (16) and (13) in (57), one reaches

$$
\begin{cases}
\frac{d}{dt}M_t = \left(\dot{\beta}(t)\hat{c}_t + \frac{\dot{\alpha}(t)}{\alpha(t)}(1-\hat{c}_t\beta(t))\right)M_t + \left(\dot{\beta}(t)-\frac{\dot{\alpha}(t)}{\alpha(t)}\beta(t)\right)m_t \\
\frac{d}{dt}Q_t^\xi = \left(\dot{\beta}(t)\hat{c}_t + \frac{\dot{\alpha}(t)}{\alpha(t)}(1-\hat{c}_t\beta(t))\right)Q_t^\xi + \left(\dot{\beta}(t)-\frac{\dot{\alpha}(t)}{\alpha(t)}\beta(t)\right)q_t^\xi \\
\frac{d}{dt}Q_t^\eta = \left(\dot{\beta}(t)\hat{c}_t + \frac{\dot{\alpha}(t)}{\alpha(t)}(1-\hat{c}_t\beta(t))\right)Q_t^\eta + \left(\dot{\beta}(t)-\frac{\dot{\alpha}(t)}{\alpha(t)}\beta(t)\right)q_t^\eta
\end{cases}, \quad (58)
$$

which recovers equation (15) of Result 3.1. Noting that $\hat{w}_t \in \text{span}(\mu, \xi, \eta)$ (see Result 2.1), the differential equation (56) becomes, for the orthogonal component $X_t^\perp \in \text{span}(\mu, \xi, \eta)^\perp$

$$
\frac{d}{dt}X_t^\perp = \left(\dot{\beta}(t)\hat{c}_t + \frac{\dot{\alpha}(t)}{\alpha(t)}(1-\hat{c}_t\beta(t))\right)X_t^\perp \quad (59)
$$

which recovers (17) of Result 3.1. This can be explicitly solved as

$$
X_t^\perp = X_0^\perp e^{\int_0^t \left(\dot{\beta}(t)\hat{c}_t + \frac{\dot{\alpha}(t)}{\alpha(t)}(1-\hat{c}_t\beta(t))\right)dt} \quad (60)
$$

Finally,

$$
\begin{aligned}
Q_t &\equiv \|X_t\|^2 \\
&= M_t^2 + n(Q_t^\xi)^2 + n\sigma^2(Q_t^\eta)^2 + \|X_t^\perp\|^2 \\
&= M_t^2 + n(Q_t^\xi)^2 + n\sigma^2(Q_t^\eta)^2 + e^{2\int_0^t \left(\dot{\beta}(t)\hat{c}_t + \frac{\dot{\alpha}(t)}{\alpha(t)}(1-\hat{c}_t\beta(t))\right)dt}
\end{aligned} \quad (61)
$$

which concludes the derivation of Result 3.1 $\qquad\square$

## B.1 Derivation of Remark 3.2

The derivation of Remark 3.2 follows identical steps, building on the observation that the discretized flow 19 is explicitly expressed as

$$
\begin{aligned}
X_{t_{k+1}} = X_{t_k} + \delta t_k \left(\dot{\beta}(t_k)\hat{c}_{t_k} + \frac{\dot{\alpha}(t_k)}{\alpha(t_k)}(1-\hat{c}_{t_k}\beta(t_k))\right)X_{t_k} \\
+ \delta t_k \left(\dot{\beta}(t_k)-\frac{\dot{\alpha}(t_k)}{\alpha(t_k)}\beta(t_k)\right)\hat{w}_{t_k}\text{sign}(\hat{w}_{t_k}^\top X_{t_k})
\end{aligned} \quad (62)
$$

Taking overlaps with $\mu, \xi, \eta$ yields

$$
\begin{cases}
\frac{\mu^\top X_{t_{k+1}}}{d} = \frac{\mu^\top X_{t_k}}{d} + \delta t_k\left(\dot{\beta}(t_k)\hat{c}_{t_k} + \frac{\dot{\alpha}(t_k)}{\alpha(t_k)}(1-\hat{c}_{t_k}\beta(t_k))\right)\frac{\mu^\top X_{t_k}}{d} + \delta t_k\left(\dot{\beta}(t_k)-\frac{\dot{\alpha}(t_k)}{\alpha(t_k)}\beta(t_k)\right)\text{sign}(\hat{w}_{t_k}^\top X_{t_k})\frac{\mu^\top \hat{w}_{t_k}}{d} \\
\frac{\xi^\top X_{t_{k+1}}}{nd} = \frac{\xi^\top X_{t_k}}{nd} + \delta t_k\left(\dot{\beta}(t_k)\hat{c}_{t_k} + \frac{\dot{\alpha}(t_k)}{\alpha(t_k)}(1-\hat{c}_{t_k}\beta(t_k))\right)\frac{\xi^\top X_{t_k}}{nd} + \delta t_k\left(\dot{\beta}(t_k)-\frac{\dot{\alpha}(t_k)}{\alpha(t_k)}\beta(t_k)\right)\text{sign}(\hat{w}_{t_k}^\top X_{t_k})\frac{\xi^\top \hat{w}_{t_k}}{nd} \\
\frac{\eta^\top X_{t_{k+1}}}{nd\sigma^2} = \frac{\eta^\top X_{t_k}}{nd\sigma^2} + \delta t_k\left(\dot{\beta}(t_k)\hat{c}_{t_k} + \frac{\dot{\alpha}(t_k)}{\alpha(t_k)}(1-\hat{c}_{t_k}\beta(t_k))\right)\frac{\eta^\top X_{t_k}}{nd\sigma^2} + \delta t_k\left(\dot{\beta}(t_k)-\frac{\dot{\alpha}(t_k)}{\alpha(t_k)}\beta(t_k)\right)\text{sign}(\hat{w}_{t_k}^\top X_{t_k})\frac{\eta^\top \hat{w}_{t_k}}{nd\sigma^2}
\end{cases} \quad (63)
$$

Like in the continuous case, one makes the heuristic assumption that $\text{sign}(\hat{w}_{t_k}^\top X_{t_k})$ stays constant along the flow, taking value $\pm 1$ with equal probability, depending on the initial condition. Doing so

yields

$$
\begin{cases}
M_{t_{k+1}} = M_{t_k} + \delta t_k \left( \dot{\beta}(t_k)\hat{c}_{t_k} + \frac{\dot{\alpha}(t_k)}{\alpha(t_k)}(1 - \hat{c}_{t_k}\beta(t_k)) \right) M_{t_k} + \delta t_k \left( \dot{\beta}(t_k) - \frac{\dot{\alpha}(t_k)}{\alpha(t_k)}\beta(t_k) \right) m_{t_k} \\
Q^\xi_{t_{k+1}} = Q^\xi_{t_k} + \delta t_k \left( \dot{\beta}(t_k)\hat{c}_{t_k} + \frac{\dot{\alpha}(t_k)}{\alpha(t_k)}(1 - \hat{c}_{t_k}\beta(t_k)) \right) Q^\xi_{t_k} + \delta t_k \left( \dot{\beta}(t_k) - \frac{\dot{\alpha}(t_k)}{\alpha(t_k)}\beta(t_k) \right) q^\xi_{t_k} \\
Q^\eta_{t_{k+1}} = Q^\eta_{t_k} + \delta t_k \left( \dot{\beta}(t_k)\hat{c}_{t_k} + \frac{\dot{\alpha}(t_k)}{\alpha(t_k)}(1 - \hat{c}_{t_k}\beta(t_k)) \right) Q^\eta_{t_k} + \delta t_k \left( \dot{\beta}(t_k) - \frac{\dot{\alpha}(t_k)}{\alpha(t_k)}\beta(t_k) \right) q^\eta_{t_k}
\end{cases},
$$
$$(64)$$

which recovers equation (20). Equation (21) follows from equation (62) and the fact that $\hat{w}_t \in \mathrm{span}(\mu, \xi, \eta)$, see Result 2.1. This recursion can be explicitly solved as

$$
X^\perp_{t_{k+1}} = X^\perp_{t_0} \prod_{\ell=0}^{k} \left( 1 + \left( \dot{\beta}(t_\ell) + \frac{\dot{\alpha}(t_\ell)}{\alpha(t_\ell)}(1 - \hat{c}_{t_\ell}\beta(t_\ell)) \right) \delta t_\ell \right)^2 .
$$
$$(65)$$

Using the fact that $\|X^\perp_{t_0}\|/d = 1$ with high probability and the definition of the summary statistics $Q$ finally yields equation (22). $\qquad\square$

## B.2 Derivation of Corollary 3.3

As implied by Result 3.1, the mean of the generated mixture is contained in $\mathrm{span}(\mu, \xi, \eta)$ and characterized by the summary statistics $M_1, Q^\eta_1, Q^\xi_1$ at time $t = 1$. Furthermore

$$
\begin{aligned}
\frac{1}{d}\|\hat{\mu} - \mu\|^2 &= \frac{1}{d}\|\hat{\mu}\|^2 - 2\frac{\hat{\mu}^\top\mu}{d} + 1 \\
&= M_1^2 + n(Q^\xi_1)^2 + n\sigma^2(Q^\eta_1)^2 - 2R_1 + 1.
\end{aligned}
$$
$$(66)$$

This recovers (23). Equation (24) follows from the definition of the cosine similarity.

The derivation of the $\Theta_n(1/n)$ decay of this distance require more work. The first step lies in the analysis of the exact flow (1).

**Remark B.1.** *(exact velocity field) For the target density $\rho_1$ (8), b is given by Efron (2011); Albergo et al. (2023) as*

$$
b(x,t) = \left( \dot{\beta}(t) - \frac{\dot{\alpha}(t)}{\alpha(t)}\beta(t) \right) \left( \frac{\beta(t)\sigma^2}{\alpha(t)^2 + \beta(t)^2\sigma^2}x + \frac{\alpha(t)^2}{\alpha(t)^2 + \beta(t)^2\sigma^2}\mu \times \tanh(\mu^\top x) \right) + \frac{\dot{\alpha}(t)}{\alpha(t)}x.
$$
$$(67)$$

The formula (67) follows from an application of Tweedie's formula Efron (2011) for the the density (8). Note that with high probability for $x \sim \rho_0$, or for any $x$ such that $\mu^\top x \gg 1$,

$$
\tanh(\mu^\top x) = \mathrm{sign}(\mu^\top x) + o_d(1).
$$
$$(68)$$

One is now in a position to characterize the exact flow (1).

**Corollary B.2.** *(Summary statistics for the exact flow) Let $X^\star_t$ be a solution of the exact flow (1) from an initialization $X^\star_0 \sim \rho_0$. Consider the summary statistic*

$$
M^\star_t \equiv \frac{\mu^\top X^\star_t}{d}.
$$
$$(69)$$

*Asymptotically, $M^\star_t$ is equal with probability $1/2$ to the solution of the differential equation*

$$
\frac{d}{dt}M^\star_t = \left( \dot{\beta}(t)c_t + \frac{\dot{\alpha}(t)}{\alpha(t)}(1 - c_t\beta(t)) \right) M^\star_t + \left( \dot{\beta}(t) - \frac{\dot{\alpha}(t)}{\alpha(t)}\beta(t) \right) \frac{\alpha(t)^2}{\alpha(t)^2 + \beta(t)^2\sigma^2}
$$
$$(70)$$

*and with probability $1/2$ to the opposite thereof. We have introduced*

$$
c_t \equiv \frac{\beta(t)\sigma^2}{\alpha(t)^2 + \beta(t)^2\sigma^2}.
$$
$$(71)$$

Corollary B.2 follows from equation (67) using a derivation identical to that of Result 3.1, presented in Appendix B, provided the heuristic assumption is made that the tanh can always be approximated by a sign (68) along the flow. To show that the learnt flow 3.1 converges to the exact flow, observe the following scalings:

**Remark B.3.** *Let $t > 0$ and $m_t, q_t^\xi, q_t^\eta$ be defined by result 2.1. Then*

$$\left| m_t - \frac{\alpha(t)^2}{\alpha(t)^2 + \beta(t)^2 \sigma^2} \right| = \Theta_n(1/n), \quad |\hat{c}_t - c_t| = \Theta_n(1/n), \quad q_t^\xi = \Theta_n(1/n), \quad q_t^\eta = \Theta_n(1/n).$$
(72)

These observations immediately imply the following asymptotics, characterizing the difference between the learnt flow (7) and the exact flow (1):

**Corollary B.4.** *(**Convergence of the learnt flow**) Let $X_t^\star$ (resp. $X_t$) be a solution of the exact flow (1) (resp. learnt flow (7)), from a common initialization $X_0 \sim \rho_0$. Define the following summary statistics:*

$$\epsilon_t^m \equiv \frac{1}{d} \mu^\top (X_t - X_t^\star), \qquad \epsilon_t^\xi \equiv \frac{1}{nd} \xi^\top (X_t - X_t^\star), \qquad \epsilon_t^\eta \equiv \frac{1}{nd\sigma^2} \eta^\top (X_t - X_t^\star)$$
(73)

*Then with high probability these statistics obey the differential equations*

$$\begin{cases} \frac{d}{dt} \epsilon_t^m = \left( \dot{\beta}(t) c_t + \frac{\dot{\alpha}(t)}{\alpha(t)} (1 - c_t \beta(t)) \right) \epsilon_t^m + \Theta_n(1/n) \\ \frac{d}{dt} \epsilon_t^\xi = \left( \dot{\beta}(t) c_t + \frac{\dot{\alpha}(t)}{\alpha(t)} (1 - c_t \beta(t)) \right) \epsilon_t^\xi + \Theta_n(1/n) \\ \frac{d}{dt} \epsilon_t^\eta = \left( \dot{\beta}(t) c_t + \frac{\dot{\alpha}(t)}{\alpha(t)} (1 - c_t \beta(t)) \right) \epsilon_t^\eta + \Theta_n(1/n) \end{cases} ,$$
(74)

*from the initial condition $\epsilon_0^{m,\xi,\eta} = 0$. Therefore at time $t = 1$*

$$\epsilon_1^m = \Theta_n(1/n), \qquad \epsilon_1^\xi = \Theta_n(1/n), \qquad \epsilon_1^\eta = \Theta_n(1/n).$$
(75)

Corollary B.4 follows from substracting the differential equations governing the *learnt* flow of Result 3.1 and the *true* flow of Corollary B.2, using the scaling derived in Remark B.3. Finally, noting that $M_1^\star = 1$ by definition of the exact flow,

$$\begin{aligned} \frac{1}{d} \|\hat{\mu} - \mu\|^2 &= \frac{1}{d} \|\epsilon_1^m \mu + \epsilon_1^\xi \xi + \epsilon_1^\eta \eta\|^2 \\ &= (\epsilon_1^m)^2 + n(\epsilon_1^\xi)^2 + n\sigma^2 (\epsilon_1^\eta)^2 + O_d(1/\sqrt{d}) \\ &= \Theta_n(1/n). \end{aligned}$$
(76)

In the last line, we used Corollary B.4. This concludes the derivation of Corollary 3.3. Fig. 2 (right) gives a PCA visualization of the convergence of the generated density $\hat{\rho}_1$ to the target density $\rho_1$ as the number available training samples $n$ accrues. □

## C  DERIVATION OF REMARK 4.1

In this appendix, we analyze the performance of the Bayes-optimal estimator of the cluster mean, defined as the estimator minimizing the average MSE knowing the train set $\mathcal{D} = \{x_0^\mu, x_1^\mu\}_{\mu=1}^n$, the clusters variance $\sigma$, but *not* the mean $\mu$. This estimator yields the information-theoretically minimal achievable MSE, and is known to be given by the mean of the posterior distribution over the estimate $w$ of the true mean $\mu$:

$$\begin{aligned} \mathbb{P}(w|\mathcal{D}, \sigma) &= e^{-\frac{1}{2}\|w\|^2} \prod_{\mu=1}^n \left[ \frac{1}{2} e^{-\frac{1}{2\sigma^2}\|x_1^\mu - w\|^2} + \frac{1}{2} e^{-\frac{1}{2\sigma^2}\|x_1^\mu + w\|^2} \right] \\ &\equiv \frac{1}{Z} e^{-\frac{1}{2\hat{\sigma}^2}\|w\|^2 + \sum_{\mu=1}^n \ln \cosh\left( \frac{w^\top x_1^\mu}{\sigma^2} \right)}, \end{aligned}$$
(77)

where

$$\hat{\sigma}^2 \equiv \frac{\sigma^2}{n + \sigma^2}.$$
(78)

We remind the reader that the prior distribution over the cluster mean is supposed to be the standard Gaussian prior $\mathcal{N}(0, \mathbb{I}_d)$. In high dimensions, statistics associated to the posterior distribution (77) are expected to concentrate. Again, it is useful to study the partition function (normalization) $Z$ to

access some key summary statistics, which will in turn provide a sharp characterization of the vector $\hat{\mu}^\star(\mathcal{D})$ extremizing the posterior $\mathbb{P}(w|\mathcal{D}, \sigma)$.

The partition function reads

$$
\begin{aligned}
Z &= \int dw e^{-\frac{1}{2\hat{\sigma}^2}\|w\|^2 + \sum\limits_{\mu=1}^{n} \ln\cosh\left(\frac{w^\top x_1^\mu}{\sigma^2}\right)} \\
&= \int dq d\hat{q} dm d\hat{m} \prod_{\mu=1}^{n} dq_\eta^\mu d\hat{q}_\eta^\mu e^{\frac{d}{2}q\hat{q} + d\sum\limits_{\mu=1}^{n} q_\eta^\mu \hat{q}_\eta^\mu + dm\hat{m}} \\
&\quad \int dw e^{-\frac{\hat{q}}{2}\|w\|^2 - \frac{1}{2\hat{\sigma}^2}\|w\|^2 - \left(\hat{m}\mu + \sum\limits_{\mu=1}^{n} \hat{q}_\eta^\mu s^\mu z^\mu\right)^\top w} e^{-d\sum\limits_{\mu=1}^{n} \ln\cosh^{1/d}\left[\frac{d}{\sigma^2}\left(m + q_\eta^\mu\right)\right]}
\end{aligned} \tag{79}
$$

As in Appendix A, we have introduced the summary statistics

$$
q_\eta^\mu \equiv s^\mu \frac{w^\top z^\mu}{d}, \qquad\qquad m = \frac{w^\top \mu}{d}. \tag{80}
$$

The integral of (79) can be evaluated using a Laplace approximation. Again, we assume the extremizer is realized at the sample-symmetric point

$$
\begin{aligned}
\forall 1 \le \mu \le n, \ q_\eta^\mu &= q_\eta, \\
\forall 1 \le \mu \le n, \ \hat{q}_\eta^\mu &= \hat{q}_\eta.
\end{aligned} \tag{81}
$$

The partition function (79) then reduces to

$$
\begin{aligned}
Z &= \int dq d\hat{q} dq_\eta d\hat{q}_\eta dm d\hat{m} e^{\frac{d}{2}q\hat{q} + dnq_\eta\hat{q}_\eta + dm\hat{m}} \\
&\quad \int dw e^{-\frac{\hat{q}}{2}\|w\|^2 - \frac{1}{2\hat{\sigma}^2}\|w\|^2 - (\hat{m}\mu + \hat{q}_\eta\eta)^\top w} e^{-d\sum\limits_{\mu=1}^{n} \ln\cosh^{1/d}\left[\frac{d}{\sigma^2}\left(m + q_\eta\right)\right]} \\
&= \int dq d\hat{q} dq_\eta d\hat{q}_\eta dm d\hat{m} e^{\frac{d}{2}q\hat{q} + dnq_\eta\hat{q}_\eta + dm\hat{m}} e^{-d\sum\limits_{\mu=1}^{n} \ln\cosh^{1/d}\left[\frac{d}{\sigma^2}\left(m + q_\eta\right)\right]} \\
&\quad \frac{1}{(1 + \hat{\sigma}^2\hat{q})^{d/2}} e^{\frac{d}{2}\frac{\hat{\sigma}^2}{1 + \hat{\sigma}^2\hat{q}}(\hat{m}^2 + n\sigma^2\hat{q}_\eta^2)}.
\end{aligned} \tag{82}
$$

Therefore $\hat{q}_\eta, q_\eta, \hat{m}, m$ must extremize the effective action

$$
\Phi = \frac{q\hat{q}}{2} + nq_\eta\hat{q}_\eta + m\hat{m} - \frac{1}{2}\ln\left(1 + \hat{\sigma}^2\hat{q}\right) + \frac{\hat{\sigma}^2}{2(1 + \hat{\sigma}^2\hat{q})}\left(\hat{m}^2 + n\sigma^2\hat{q}_\eta^2\right) + \frac{n}{\sigma^2}|m + q_\eta|, \tag{83}
$$

leading to

$$
\begin{cases} \hat{q}_\eta = -\frac{1}{\sigma^2} \\ \hat{m} = -\frac{n}{\sigma^2} \end{cases}, \qquad\qquad \begin{cases} q_\eta = -\hat{q}_\eta\sigma^2\hat{\sigma}^2 = \frac{\sigma^2}{n + \sigma^2} \\ m = -\hat{m}\hat{\sigma}^2 = \frac{n}{n + \sigma^2} \end{cases}. \tag{84}
$$

Refining $\sigma^2 q_\eta \leftarrow q_\eta$ so that

$$
q_\eta \equiv \frac{w^\top \eta}{nd\sigma^2}, \tag{85}
$$

as in Remark 4.1, one finally reaches

$$
q_\eta = \frac{1}{n + \sigma^2}, \qquad\qquad m = \frac{n}{n + \sigma^2}, \tag{86}
$$

Thus, remembering $\hat{\mu}^\star(\mathcal{D}) = \langle w \rangle$ (where the bracket notation denotes averages with respect to the posterior $P(\cdot|\mathcal{D}, \sigma)$:

$$
\frac{\mu^\top \hat{\mu}^\star(\mathcal{D})}{d} = \left\langle \frac{w^\top \mu}{d} \right\rangle = \frac{n}{n + \sigma^2}, \qquad\qquad \frac{\eta^\top \hat{\mu}^\star(\mathcal{D})}{nd\sigma^2} = \left\langle \frac{w^\top \eta}{nd\sigma^2} \right\rangle = \frac{1}{n + \sigma^2}, \tag{87}
$$

using the concentration of the bracketed quantities. Furthermore,

$$\frac{\|\hat{\mu}^\star(\mathcal{D})\|^2}{d} = \frac{1}{d}\|\langle w\rangle\|^2 = \frac{\mu^\top\langle w\rangle}{d} = m. \tag{88}$$

We employed the Nishimori identity (Nishimori, 2001; Iba, 1999).Note further the identity:

$$\frac{\|\hat{\mu}^\star(\mathcal{D})\|^2}{d} = m = m^2 + n\sigma^2 q_\eta^2 = \frac{1}{d}\|m\mu + q_\eta\eta\|^2, \tag{89}$$

which implies that the norm of $\hat{\mu}^\star(\mathcal{D})$ is equal to the norm of its projection in $\mathrm{span}(\mu, \eta)$, which means that asymptotically the former is contained in the latter. One is now in a position to derive the Bayes-optimal MSE of Remark 4.1. With high probability

$$\frac{1}{d}\|\hat{\mu}^\star(\mathcal{D}) - \mu\|^2 = m + 1 - 2m = 1 - m = \frac{\sigma^2}{n + \sigma^2}. \tag{90}$$

This completes the derivation of Remark 4.1 $\qquad\qquad\square$

# D   FURTHER SETTINGS

## D.1   IMBALANCED CLUSTERS

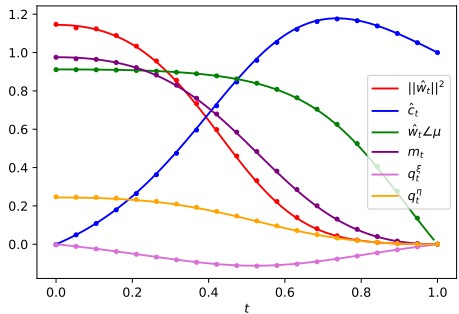

Figure 5: $n = 4, \sigma = 0.9, \lambda = 0.1, \alpha(t) = 1 - t, \beta(t) = t$. Imbalanced mixture with relative weights $\rho = 0.24$ and $1 - \rho = 0.76$. Solid lines: theoretical predictions of Result 2.1: squared norm of the DAE weight vector $\|\hat{w}_t\|^2$ (red), skip connection strength $\hat{c}_t$ (blue) cosine similarity between the weight vector $\hat{w}_t$ and the target cluster mean $\mu$, $\hat{w}_t \angle \mu \equiv \hat{w}_t^\top \mu / \|\mu\|\|\hat{w}_t\|$ (green), components $m_t, q_t^\xi, q_t^\eta$ of $\hat{w}_t$ along the vectors $\mu, \xi, \eta$ (purple, pink, orange). Dots: numerical simulations in $d = 5 \times 10^4$, corresponding to training the DAE (9) on the risk (10) uisng the `Pytorch` implementation of full-batch Adam, with learning rate $0.001$ over $20000$ epochs and weight decay $\lambda = 0.1$. The experimental points correspond to a single instance of the model.

In this appendix, we address the case of a binary homoscedastic but *imbalanced* mixture

$$\rho_1 = \rho\mathcal{N}(\mu, \sigma^2\mathbb{I}_d) + (1 - \rho)\mathcal{N}(-\mu, \sigma^2\mathbb{I}_d), \tag{91}$$

where $\rho \in (0, 1)$ controls the relative weights of the two clusters. The target density considered in the main text (8) thus corresponds to the special case $\rho = 1/2$.

It is immediate to verify that the derivations presented in Appendices A, B carry through. In other words, Result 2.1, Result 3.1 and Corollary 3.3 still exactly hold. Figure 5 shows that the sharp characterization of Result 2.1 indeed still tightly captures the learning curves of a DAE, trained on *imbalanced* clusters, using the `Pytorch` (Paszke et al., 2019) implementation of the Adam (Kingma & Ba, 2014) optimizer.

An important consequence of this observation is that the generative model will generate a *balanced* density $\hat{\rho}_1$, failing to capture the asymmetry of the target distribution $\rho_1$. This echoes the findings of Biroli & Mézard (2023) in the related setting of a target ferromagnetic Curie-Weiss model, where they argue that the asymmetry of the ground state can only be learnt for $n \gg d$ samples.

## D.2 DAE WITHOUT SKIP CONNECTION

In this appendix, we examine the importance of the skip connection in the DAE architecture (9). More precisely, we consider the generative model parameterized by the DAE *without* skip connection

$$g_{w_t}(x) = w_t \varphi(w_t^\top x) \tag{92}$$

where $\varphi$ is an activation admitting horizontal asymptots at $+1$ $(-1)$ in $+\infty$ $(-\infty)$. A tight characterization of the learnt weight $\hat{w}_t$ can also straightfowardly be accessed, and is summarized in the following result, which is the equivalent of Result 2.1 for the DAE without skip connection (92)

**Result D.1.** *(**Sharp characterization of the trained weight of (92)**) For any given activation $\varphi$ satisfying $\varphi(x) \xrightarrow{x \to \pm\infty} \pm 1$ and any $t \in [0, 1]$, in the limit $d \to \infty$, $n$, $\|\boldsymbol{\mu}\|^2/d$, $\sigma = \Theta_d(1)$, the learnt weight vector $\hat{\boldsymbol{w}}_t$ of the DAE without skip connection (92) trained on the loss (10) is asymptotically contained in $\mathrm{span}(\boldsymbol{\mu}, \boldsymbol{\eta})$ (in the sense that its projection on the orthogonal space $\mathrm{span}(\boldsymbol{\mu}, \boldsymbol{\eta})^\perp$ has asymptotically vanishing norm). The components of $\hat{\boldsymbol{w}}_t$ along each of these two vectors is given by the summary statistics*

$$m_t = \frac{\boldsymbol{\mu}^\top \hat{\boldsymbol{w}}_t}{d}, \qquad\qquad q_t^\eta = \frac{\hat{\boldsymbol{w}}_t^\top \boldsymbol{\eta}}{nd\sigma^2}, \tag{93}$$

*which concentrate as $d \to \infty$ to the time-constant quantities characterized by the closed-form formulae*

$$m = \frac{n}{\lambda + n}, \qquad\qquad q^\eta = \frac{1}{\lambda + n}. \tag{94}$$

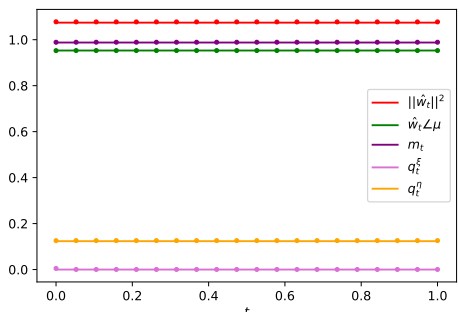

Figure 6: $n = 4, \sigma = 0.9, \lambda = 0.1, \alpha(t) = 1 - t, \beta(t) = t$. Solid lines: theoretical predictions of Result 2.1: squared norm of the weight vector $\|\hat{\boldsymbol{w}}_t\|^2$ of the DAE without skip connection (92) (red), skip connection strength $\hat{c}_t$ (blue) cosine similarity between the weight vector $\hat{\boldsymbol{w}}_t$ and the target cluster mean $\boldsymbol{\mu}$, $\hat{\boldsymbol{w}}_t \angle \boldsymbol{\mu} \equiv \hat{\boldsymbol{w}}_t^\top \boldsymbol{\mu}/\|\boldsymbol{\mu}\|\|\hat{\boldsymbol{w}}_t\|$ (green), components $m_t, q_t^\xi, q_t^\eta$ of $\hat{\boldsymbol{w}}_t$ along the vectors $\boldsymbol{\mu}, \boldsymbol{\xi}, \boldsymbol{\eta}$ (purple, pink, orange). Dots: numerical simulations in $d = 5 \times 10^4$, corresponding to training the DAE without skip connection (92) on the risk (10) uisng the `Pytorch` implementation of full-batch Adam, with learning rate $0.001$ over $20000$ epochs and weight decay $\lambda = 0.1$. The experimental points correspond to a single instance of the model.

Result D.1 follows from a straightforward adapatation of the derivation of Result 2.1 as presented in Appendix A. In fact, it naturally corresponds to setting the skip connection strength $c$ to 0 in the expression of the log partition function (41). equation (93) corresponds to the zero-gradient condition thereof.

A striking consequence of Result D.1 is that asymptotically the trained vector $\hat{w}$ of the DAE (92) *does not depend on the time $t$*. Fig. 6 provides further support of this fact, as the summary statistics measured in simulations – training the DAE (92) using the `Pytorch` implementation of full-batch Adam – are also observed to be constant in time, and furthermore to agree well with the theoretical prediction. As for the analysis presented in the main text, it is possible to track the generative flow with a finite set of summary statistics. This is the object of the following result:

**Result D.2.** *(**Summary statistics for the no-skip connection case**) Le $X_t$ be a solution of the ordinary differential equation (7) with initial condition $X_0$, when parametrized by the DAE without skip connection (92). For a given t, the projection of $X_t$ on $\mathrm{span}(\boldsymbol{\mu}, , \boldsymbol{\eta})$ is characterized by the summary statistics*

$$M_t \equiv \frac{\boldsymbol{X}_t^\top \boldsymbol{\mu}}{d}, \qquad\qquad Q_t^\eta \equiv \frac{\boldsymbol{X}_t^\top \boldsymbol{\eta}}{nd\sigma^2}. \qquad (95)$$

*With probability asymptotically $1/2$ the summary statistics $M_t, Q_t^\eta$ (15) concentrate for all t to the solution of the ordinary differential equations*

$$\begin{cases} \frac{d}{dt} M_t = \frac{\dot{\alpha}(t)}{\alpha(t)} M_t + \left( \dot{\beta}(t) - \frac{\dot{\alpha}(t)}{\alpha(t)} \beta(t) \right) \frac{n}{\lambda+n} \\ \frac{d}{dt} Q_t^\eta = \frac{\dot{\alpha}(t)}{\alpha(t)} Q_t^\eta + \left( \dot{\beta}(t) - \frac{\dot{\alpha}(t)}{\alpha(t)} \beta(t) \right) \frac{1}{\lambda+n} \end{cases}. \qquad (96)$$

The derivation of Result D.2 can be made along the exact same lines as the one for Result 3.1, presented in Appendix B. An important observation is that the flows (96) are actually the *exact* flows corresponding to a particular Gaussian mixture, as explicited in the following remark:

**Remark D.3.** *(**Generated density**) The summary statistics evolution (96) are the same evolutions that would follow from the velocity field*

$$b(x, t) = \left( \dot{\beta}(t) - \frac{\dot{\alpha}(t)}{\alpha(t)} \beta(t) \right) \left( \hat{\mu} \times \tanh\left( \hat{\mu}^\top x \right) \right) + \frac{\dot{\alpha}(t)}{\alpha(t)} x. \qquad (97)$$

*where*

$$\hat{\mu} \equiv \frac{n}{\lambda+n} \mu + \frac{1}{\lambda+n} \eta. \qquad (98)$$

*Comparing with equation (67), this is the exact velocity field associated to the singular Gaussian mixture*

$$\hat{\rho}_1(x) = \frac{1}{2} \delta(x - \hat{\mu}) + \frac{1}{2} \delta(x + \hat{\mu}). \qquad (99)$$

The generative model parameterized by the DAE *without* skip connection (92) thus learns a singular density, which is a sum of two Dirac atoms, centered at $\pm\hat{\mu}$. It thus fails to generate a good approximation of the target $\rho_1$ (8). Note however that interestingly $\hat{\mu}$ remains a good approximation of the true mean $\mu$, and actually converges thereto as $n \to \infty$. This is made more precise by the following result:

**Remark D.4.** *(**mse in the no-skip-connection case**) Let $\hat{\mu}$ be the cluster mean of the estimated density $\hat{\rho}_1$, as defined in Remark D.3. Then its squared distance to the true mean $\mu$ is*

$$\frac{1}{d} \|\hat{\mu} - \mu\|^2 = \frac{\lambda^2 + n\sigma^2}{(\lambda + n)^2} \qquad (100)$$

*The minimum is achieved for $\lambda = \sigma^2$ and is equal to the Bayes MSE 4.1. In particular, this MSE decays as $\Theta_n(1/n)$.*

Strikingly, the generative model parametrized by (92) manages to achieve the Bayes optimal $\Theta_n(1/n)$ rate in terms of the *estimation MSE* over the cluster means, but completely fails to accurately estimate the true variance.

## E  A FULLY EXPRESSIVE MODEL MEMORIZES

In this appendix, we show that the absence of memorization – defined as the ability of the generative model to generate new samples, and not just retrieve the training samples– is enabled by the network parametrization of the generative model. In fact, a fully expressive (flexible) model would in fact *memorize* the train set. Consider the network-parametrized minimization problem over the parameter space $\{\theta_t\}_{t\in[0,1]}$

$$\hat{\mathcal{R}}(\{\theta_t\}_{t\in[0,1]}) = \frac{1}{n} \int_0^1 \sum_{\mu=1}^n \mathbb{E}_{x_0} \|\boldsymbol{f}_{\theta_t}(\boldsymbol{x}_t^\mu) - \boldsymbol{x}_1^\mu\|^2 \, dt. \qquad (101)$$

For ease of discussion, compared to equation (5), we consider the case where for each sample $x_1^\mu$ of the target $\rho_1$, we sample an infinity of noises $x_0$ from the easy-to-sample base Gaussian distribution $\rho_0$, which corresponds to averaging over $x_0$ in equation (101). Note that doing so, compared to the case where only one $x_0^\mu$ is sampled for very $x_1^\mu$, is expected to prevent the model from overfitting the noise and should only improve the performance. Now consider replacing the minimization equation (101) by the minimization over the space of *all* denoising functions

$$\hat{\mathcal{R}}[f] = \frac{1}{n} \int_0^1 \sum_{\mu=1}^n \mathbb{E}_{x_0} \|\boldsymbol{f}(\boldsymbol{x}_t^\mu, t) - \boldsymbol{x}_1^\mu\|^2 \, dt = \int_0^1 \mathbb{E}_{x_1 \sim \tilde{\rho}_1} \mathbb{E}_{x_0} \|\boldsymbol{f}(\boldsymbol{x}_t, t) - \boldsymbol{x}_1\|^2 \, dt. \tag{102}$$

In equation (102) we denoted $\tilde{\rho}_1$ the empirical distribution supported on the training samples

$$\tilde{\rho}_1(x_1) = \frac{1}{n} \sum_{\mu=1}^n \delta(x_1 - x_1^\mu) \tag{103}$$

and remind that the distribution of the variable $x_t$ follows from its definition as $x_t = \alpha(t)x_0 + \beta(t)x_1$. Finally, in equation (102), instead of minimizing a function $f_t : \mathbb{R}^d \to \mathbb{R}^d$ for each $t \in [0,1]$, we have without loss of generality rewritten $f_t(\cdot) = f(\cdot, t)$. The objective equation (102) can be seen as the limit of equation (5) when the network is infinitely flexible and can express *any* denoising function $f$. Comparing equation (102) to equation (4), it follows from Albergo et al. (2023) that the minimizer of equation (102) leads to a flow mapping the base Gaussian distribution $\rho_0$ to the density $\tilde{\rho}_1$ equation (103), since equation (102) is the *population* objective for $\tilde{\rho}_1$. Since $\rho_0$ and $\tilde{\rho}_1$ are both Gaussian mixtures (with the clusters of the latter being of vanishing variance), the corresponding velocity field is furthermore explicitly given in Appendix A of Albergo et al. (2023). Therefore, an infinitely expressive model minimizing the empirical risk equation (101) leads to a generated density $\tilde{\rho}_1$. In other words, it only allows to generate samples $x_1^\mu$ from the training set, and the generated mixture has $n$ clusters with $0$ variance. In contrast, the DAE-parametrized model equation (9) learns a bimodal mixture with non-zero variance.

## F  WASSERSETEIN DISTANCE

In this appendix, we derive a precise description of the generated distribution $\hat{\rho}_1$ and the target $\rho_1$. We remind that the distribution of its projection in $\text{span}(\xi, \mu_{\text{emp.}})^\perp$ follows the Gaussian distribution

$$X_1^\perp \sim \mathcal{N}\left(0, \underbrace{e^{2\int_0^1\left(\dot{\beta}(t)\hat{c}_t + \frac{\dot{\alpha}(t)}{\alpha(t)}(1-\hat{c}_t\beta(t))\right)dt}}_{\hat{\sigma}^2} \mathbb{I}_{d-2}\right). \tag{104}$$

Observe that from Result 2.1,

$$\hat{c}_t = \frac{\sigma^2 \beta(t)}{\alpha(t)^2 + \beta(t)^2 \sigma^2} + \Theta_n(1/n). \tag{105}$$

Thus,

$$\ln \hat{\sigma}^2 = 2 \int_0^1 \frac{\dot{\alpha}(t)\alpha(t) + \dot{\beta}(t)\beta(t)\sigma^2}{\alpha(t)^2 + \beta(t)^2\sigma^2} + \Theta_n(1/n) = [\ln\left(\alpha(t)^2 + \beta(t)^2\sigma^2\right)]_0^1 + \Theta_n(1/n) = \ln \sigma^2 + \Theta_n(1/n). \tag{106}$$

Thus

$$\hat{\sigma}^2 = \sigma^2 + \Theta_n(1/n) \tag{107}$$

We are now in a position to compute the Mixture Wasserstein distance between the generated density $\hat{\rho}_1$ and the target $\rho_1$. Because these are $d-$ dimensional distributions, we normalize this distance by $1/d$ so as to have order 1 metrics in the considered asymptotic limit $d \to \infty$. Because of this, the precise distribution of the clusters of $\hat{\rho}_1$ in the two dimensional space $\text{span}(\xi, \mu_{\text{emp.}})$ does *not* matter, provided it does not involve moments diverging with $d$. We will show that this very reasonable assumption is indeed verified, after the computation of the distance.

### F.1 WASSERSTEIN DISTANCE

We aim at evaluating a distance metric in the space of distributions to quantify the discrepancy between the true density $\rho_1$ and the generated $\hat{\rho}_1$. Many natural metrics (e.g. the KL divergence) are however intractable in our setting. We take inspiration from the Gaussian mixture Wasserstein distance $MW_2$, proposed in Delon & Desolneux (2020) as variant of the $W_2$ distance for Gaussian mixtures. Because $\hat{\rho}_1$ is a mixture, but the clusters in $\mathrm{span}(\xi, \mu_{\mathrm{emp.}})$ are only halves of Gaussian clusters, we define and employ a very similar metric for arbitrary mixtures:

**Definition F.1.** (*mixture Wasserstein distance*) Given two mixtures $\sum_{i=1}^{K} \rho_i \mu_i$ and $\sum_{i=1}^{J} \tau_i \nu_i$ (with $\mu_i, \nu_i$ not necessarily Gaussian densities), the $MW_2$ distance is defined as

$$MW_2^2 = \min_{w \in \mathbb{R}^{K \times J} | w 1_J = (\rho_1, \ldots, \rho_K), w^\top 1_K = (\tau_1, \ldots, \tau_J)} \frac{1}{d} \sum_{k=1}^{K} \sum_{j=1}^{J} w_{kj} \mathcal{W}_2^2(\mu_k, \nu_j). \tag{108}$$

This is the same definition as Delon & Desolneux (2020), except that we allow for non-Gaussian $\mu_i, \nu_i$. Note that we introduced without loss of generality a normalization $1/d$, since we are comparing $d-$dimensional densities, and expect the distance to scale with $d$. With the normalization, the metric stays $\Theta_d(1)$ as $d \to \infty$. In the present setting, this evaluates to

$$MW_2^2[\rho_1, \hat{\rho}_1] = \frac{1}{d} \mathcal{W}_2^2(\hat{\rho}_1^+, \mathcal{N}(\mu, \sigma^2)) + \frac{1}{d} \mathcal{W}_2^2(\hat{\rho}_1^-, \mathcal{N}(-\mu, \sigma^2)) \tag{109}$$

where we introduced the densities $\hat{\rho}_1 = 1/2 \hat{\rho}_1^+ + 1/2 \hat{\rho}_1^-$ for the two clusters of $\hat{\rho}_1$ centered at $\pm\hat{\mu}$. We denote further decompose $\rho_1^\pm = \rho_1^{\pm\|} \otimes \rho_1^{\pm\perp}$ into the product of the distribution $\rho_1^{\pm\|}$ in $\mathrm{span}(\xi, \mu_{\mathrm{emp.}})$ and the Gaussian $d - 2$ dimensional density $\rho_1^{\pm\perp}$ in $\mathrm{span}(\xi, \mu_{\mathrm{emp.}})^\perp$. We can similarly decompose the target Gaussian density $\mathcal{N}(\pm\mu, \sigma^2 \mathbb{I}_d) = \mathcal{N}(\pm\mu, \sigma^2 \mathbb{I}_2) \otimes \mathcal{N}(0, \sigma^2 \mathbb{I}_{d-2})$. Using the the properties of Wasserstein distances between product measures Panaretos & Zemel (2019),

$$\frac{1}{d} \mathcal{W}_2^2(\hat{\rho}_1^+, \mathcal{N}(\mu, \sigma^2)) = \frac{1}{d} \mathcal{W}_2^2(\rho_1^{+\|}, \mathcal{N}(\pm\mu^\|, \sigma^2)) + \frac{1}{d} \mathcal{W}_2^2(\rho_1^{+\perp}, \mathcal{N}(0, \sigma^2)) \tag{110}$$

Note that since $\rho_1^{\pm\perp}$ is Gaussian with variance $\hat{\sigma}^2$, the second term corresponds to the Wasserstein distance between two Gaussian distributions and read

$$\frac{1}{d} \mathcal{W}_2^2(\rho_1^{+\perp}, \mathcal{N}(0, \sigma^2)) = (\sigma - \hat{\sigma})^2 = \Theta_n(1/n). \tag{111}$$

We now bound $\frac{1}{d} \mathcal{W}_2^2(\rho_1^{+\|}, \mathcal{N}(\pm\mu^\|, \sigma^2))$. Note that the two densities are centered around $\hat{\mu}$ and $\mu$. The discrepancy between these means will provide the dominant term in the distance. To see this, we introduce $\nu_1(x) = \delta(x - \hat{\mu})$ and $\nu_2(x) = \delta(x - \mu)$, two Diracs centered at the means, and upper-bound $\frac{1}{d} \mathcal{W}_2^2(\rho_1^{+\|}, \mathcal{N}(\pm\mu^\|, \sigma^2))$ using the triangular inequality

$$\frac{1}{d} \mathcal{W}_2^2(\rho_1^{+\|}, \mathcal{N}(\pm\mu^\|, \sigma^2)) \leq \frac{1}{d} \mathcal{W}_2^2(\rho_1^{+\|}, \nu_1) + \frac{1}{d} \mathcal{W}_2^2(\nu_1, \nu_2) + \frac{1}{d} \mathcal{W}_2^2(\nu_2, \mathcal{N}(\pm\mu^\|, \sigma^2)). \tag{112}$$

The last term is asymptotically vanishing as $\Theta_d(1/d)$. Under very mild assumption on $\rho_1^{+\|}$ (which we show are verified in the next subsection), the Wasserstein distance between two-dimensional distributions $\mathcal{W}_2^2(\rho_1^{+\|}, \nu_1)$ should be $\Theta_d(1)$, so the first term also vanishes as $\Theta_d(1/d)$. The second term is equal to

$$\frac{1}{d} \mathcal{W}_2^2(\nu_1, \nu_2) = \frac{1}{d} \|\mu - \hat{\mu}\|^2 = \Theta_n(1/n), \tag{113}$$

using result B.3. The derivation proceeds identically for the other pair of clusters $\frac{1}{d} \mathcal{W}_2^2(\hat{\rho}_1^-, \mathcal{N}(-\mu, \sigma^2))$. Putting everything together, we reach

$$MW_2^2[\rho_1, \hat{\rho}_1] \leq \Theta_n(1/n) \tag{114}$$

**Bayes optimal rate**   We briefly consider the mixture corresponding to the bimodal mixture centered at the Bayes optimal mean estimator $\pm\hat{\mu}(\mathcal{D})$, and assuming perfect knowledge of the cluster covariances. Again, this is provided as an insightful baseline, and does *not* constitute a generative model, since exact oracle knowledge of the form of $\rho_1$ and of $\sigma^2$ is assumed. For the Bayes estimator:

$$MW_2^2[\rho_1, \hat{\rho}_1] = \frac{2}{d}\|\mu - \hat{\mu}\|^2 = \Theta_n(1/n), \tag{115}$$

using Result 4.1.

We now briefly give two other examples for generative models differently parametrized, for which the generated density does not converge to the target $\rho_1$ in $MW_2$ distance.

**Auto-encoder without skip connection**   As derived in D, the generated density when the model is parametrized by a DAE *without* skip connection is a degenerate mixture $1/2\delta(\cdot - \hat{\mu}) + 1/2\delta(\cdot + \hat{\mu})$, which corresponds to setting $\hat{\sigma} = 0$ in the above derivation. Thus, it follows that

$$MW_2^2[\rho_1, \hat{\rho}_1] = \Theta_n(1), \tag{116}$$

i.e. without skip connection the generative model fails to learn to generate the target mixture.

**Fully expressive model**   We now consider the case of a model which memorizes the train set, as discussed in Appendix E. In this case

$$\hat{\rho}_1(x) = \frac{1}{n}\sum_{\mu=1}^n \delta(x - x_1^\mu), \tag{117}$$

which is a (degenerate) Gaussian mixture. It is straightforward to see that for any $x_1^\mu$, $\mathcal{W}_2^2[\mathcal{N}(\pm\mu, \sigma^2\mathbb{I}_d, \delta(\cdot - x_1^\mu)] \geq \sigma^2 = \Theta_n(1)$, and therefore

$$MW_2^2[\rho_1, \hat{\rho}_1] \geq \sigma^2 = \Theta_n(1). \tag{118}$$

Thus $\hat{\rho}_1$ is bounded away from the target $\rho_1$.

We close the appendix by deriving the precise form of $\hat{\rho}_1^{\pm\|}$, although the precise distribution in this two-dimensional space is asymptotically irrelevant for all the considered metrics, as we showed.

### F.2   Distribution in $\text{span}(\xi, \mu_{\text{emp.}})$

We study in more detail the dynamics of $X_t^\|$, defined as the projection of $X_t$ in $\text{span}(\xi, \mu_{\text{emp.}})$. Since the initial $X_0 \sim \rho_0$ is Gaussian, so is its projection $X_0^\| \sim \mathcal{N}(0, \mathbb{I}_2)$. Let us also call $\hat{w}_t^\|$ the projection of $\hat{w}_t$ in $\text{span}(\xi, \mu_{\text{emp.}})$. Projecting the dynamics equation (7) into $\text{span}(\xi, \mu_{\text{emp.}})$,

$$\dot{X}_t^\| = \left(\dot{\beta}(t)\hat{c}_t + \frac{\dot{\alpha}(t)}{\alpha(t)}(1 - \hat{c}_t\beta(t))\right)X_t^\| \pm \left(\dot{\beta}(t) - \frac{\dot{\alpha}(t)}{\alpha(t)}\beta(t)\right)\hat{w}_t^\|, \tag{119}$$

where the sign of the drift term is given by $\text{sign}(X_0^{\|\top}\hat{w}_0^\|)$. Like in Appendix B, we assumed that $\text{sign}(\hat{w}_t^\top X_t)$ stays constant during the transport. This can be solved in closed form for $t = 1$ as

$$X_1^\| = X_0^\| e^{\int_0^1 \gamma(t)dt} + \text{sign}(X_0^{\|\top}\hat{w}_0^\|)e^{\int_0^1 \gamma(t)dt}\int_0^1 e^{-\int_0^t \gamma(s)ds}\left(\dot{\beta}(t) - \frac{\dot{\alpha}(t)}{\alpha(t)}\beta(t)\right)\hat{w}_t^\| dt \tag{120}$$

where we used the shorthand

$$\gamma(t) \equiv \left(\dot{\beta}(t)\hat{c}_t + \frac{\dot{\alpha}(t)}{\alpha(t)}(1 - \hat{c}_t\beta(t))\right). \tag{121}$$

The second term multiplied by the sign corresponds to $\hat{\mu} \in \text{span}(\xi, \mu_{\text{emp.}})$ as characterized by Result 3.1. The distribution of $X_1^\|$ follows from that of the Gaussian $X_0^\|$. If $X_0^\|$ is in the half-space $\{x \in \mathbb{R}^2 | x^\top \hat{w}_0^\| \geq 0\}$ then $X_1^\| = \hat{\sigma}X_0^\| + \hat{\mu}$; If $X_0^\|$ is in the half-space $\{x \in \mathbb{R}^2 | x^\top \hat{w}_0^\| \leq 0\}$ then $X_1^\| = \hat{\sigma}X_0^\| - \hat{\mu}$. In other words, the distribution of $X_1^\perp$ is a mixture of two clusters, at $\pm\hat{\mu}$. Each cluster corresponds to half a Gaussian cluster of variance $\hat{\sigma}^2\mathbb{I}$, i.e. a Gaussian cluster cleft along a hyperplane whose othogonal vector is $\hat{w}_0^\|$ as characterized by Result 2.1.

