# OpenReview forum: "Analysis of Learning a Flow-based Generative Model from Limited Sample Complexity"
_ICLR.cc/2024/Conference — ICLR 2024 poster_

### Official Review · Reviewer_AsnX · 2023-10-24

**Soundness:** 3 good
**Presentation:** 3 good
**Contribution:** 3 good
**Rating:** 8
**Confidence:** 4

**Summary:**

The paper studies a certain asymptotic limit of a flow-based generative model that uses weight-tied fully connected network with skip connection to approximate the velocity field. In such scenario, under the isotropic gaussian mixture assumption the authors provide a characterization of training dynamics in the finite sample complexity regime. Notably, the resulting asymptotic behaviour of the learned cluster means enjoys the Bayes optimal rate of $O(1/n)$.

**Strengths:**

- a complete characterization of training dynamics of shallow generative flow under the isotropic mixture of gaussians assumption
- optimality of the resulting sample asymptotics
- a neat symmetric ansatz

**Weaknesses:**

- lack of the correlation structure in the input data, which draws the conclusions to be less practical
- minor: the approach is still an ansatz

**Questions:**

N/A

---

> ### Author Response · Authors · 2023-11-17
>
> We thank the reviewer for their comments and questions, which we address below.
>
> >lack of the correlation structure in the input data, which draws the conclusions to be less practical
>
> We believe that ideas similar to the ones used in the present work could be leveraged to address mixtures with structured clusters, see also the answer to reviewer wFJz. Such an extension would allow to analyze more realistic data distributions, and potentially quantitatively capture the learning curves of some simple real datasets, like in (Cui & Zdeborová, 23). This constitutes an interesting research direction which we will further pursue in future work.

---

### Official Review · Reviewer_wFJz · 2023-10-25

**Soundness:** 3 good
**Presentation:** 3 good
**Contribution:** 3 good
**Rating:** 8
**Confidence:** 4

**Summary:**

The authors analyze a generative flow-based model to sample from a mixture of Gaussians, where at each timestep the vector field is parameterized by a two-layer neural network.
They consider a high dimensional finite sample regime with the number of samples scaling linearly in the dimension.
Using tools from statistical physics they derive a precise characterization of the optimal performance.
Their experiments corroborate the theoretical findings.

**Strengths:**

- The strongest point is the exact characterization of all important quantities.
- Clear and concise presentation of the results
- Solid experiments demonstrating the validity of the theoretical results

**Weaknesses:**

The approach seems to heavily rely on the gaussianity of the target distribution

**Questions:**

- Do you think this the same method could be used for more complex distributions, if yes what would be a concrete example?

---

> ### Author Response · Authors · 2023-11-17
>
> We thank the reviewer for their appreciation of our work, and address below their concerns.
>
> >Do you think this the same method could be used for more complex distributions, if yes what would be a concrete example?
>
> While the method indeed makes use of the fact that the target density is a Gaussian mixture, we believe that very similar ideas can be employed to address Gaussian mixture densities with structured covariances. This would provide a gateway to modeling more realistic setups, since as shown in (Cui & Zdeborová, 23), at the level of the learning process, the performance of the DAE on MNIST is quantitatively captured when modeling the latter by the matching Gaussian mixture density. Another natural extension which could be addressed with similar methods is the case of more clusters, and using an autoencoder with more hidden units. We leave the exploration of these exciting questions to future work.

---

### Official Review · Reviewer_v6CB · 2023-10-26

**Soundness:** 2 fair
**Presentation:** 3 good
**Contribution:** 2 fair
**Rating:** 3
**Confidence:** 5

**Summary:**

The paper in detail analyses the case of training a stochastic interpolation model on few samples from a bimodal Gaussian mixture model. All theoretic predictions are accompanied by experiments.

The first result 2.1 characterizes the solution that is obtained when training on a bimodal Gaussian mixture. It shows that the weight vector is contained in the span of $\mu$ (the displacement vector for the mixture components), $\xi$ (the mean of all latent points), and $\eta$ (the average offset of the data from the corresponding mean) and thus no other directions are relevant.

The second result 3.1 derives the resulting generative ODE and summarizes it in terms of the relevant space from result 2.1.

The third result 3.2 spells out the Euler integration of result 3.1.

The final result 3.3 shows that the distance and angle between the true $\mu$ and the estimated $\hat\mu$ reduce by $\Theta(1/n)$. This is the same as the Bayes optimal rate.

**Strengths:**

The paper considers an interesting question, how generative models generalize as a function of the number of training samples.

The solution for the bimodal Gaussian Mixture model appears sound and plausible.

All theoretical results are accompanied by experiments closely matching the prediction, increasing the credibility of the theoretical results.

**Weaknesses:**

I find the main technical result presented in a misleading way: Results 2.1 and 3.1 show that the weight vector and the ODE dynamic are not orthogonal to $\mu$. I think a better way to describe the behavior of the system would be to span the relevant space via $\eta$ and $\hat\mu_+$ and $\hat\mu_-$, corresponding to the empirical mean of the samples at $\pm\mu$. I assume that this is also sufficient to span the weight vector (please correct me if I’m wrong), and it would make clear that the model only has access to the empirical means. The remaining results could then be adapted to show how the sampling process is able to reproduce the empirical means of the two modes. Then, the Bayes-optimal baseline could be inserted to show that the empirical estimates go down with Theta(1/n), and that the flow is able to achieve the same rate.

I also find that the research question formulated in the beginning is not really addressed in the main text, whether the network architecture determines whether a generative model memorizes training data. If my above understanding is correct, then the model actually does learn the training data by heart (i.e. it predicts the empirical means), and the rate to the true solution is essentially given how fast empirical means converge to the mean of the generative distribution. Also, the rate $\Theta(1/n)$ is not affected by the regularization $\lambda$, and the network architecture is not varied so as to judge wether this particular setup has a particular convergence rate.

Minor points:
- Formatting of contributions via itemize
- Sentence? „Note that (9) is a special case of the architecture studied in Cui & Zdeborová (2023), and differs from the very similar network considered in Shah et al. (2023) in its slightly more general activation (Shah et al. (2023) address the case φ = tanh)“
- Result 3.1 Le X_t -> Let X_t
- finding that that the DAE on p. 9

**Questions:**

1. How many dimensions are needed to span $\hat w_t$ for all $t$? From a simple drawing of $\eta, \mu_+, \mu_-$ I conjecture that two dimensions are enough (similar to first weakness).

2. Why is $\mu \propto d$ a reasonable scaling? This seems like an unrealistic choice to me. In practice, data is often normalized say to a fixed range $[-1, 1]$ per dimension, so in order to obtain the scaling behavior the means have to be $\mu = (\pm 1, …, \pm 1)$, i.e. both mixture components are at the corners of the hypercube. Alternative question: Does the sample complexity also transfer to $\|\mu\|^2 < O(d)$, e.g. $O(1)$? I would guess that the other directions start playing a role then.

3. What is the solution to this simple setup intuitively? Can you provide a simple drawing of $0, \eta, \mu_+, \mu_-, \mu, \eta$ and a learnt trajectory? If the required dimension is indeed two, this should be easy.

4. What is the shape of the learnt distribution within the two clusters, i.e. what is the local density in each cluster?

5. Fig. 2: Why are the learnt means biased in one direction? Is more training data added sequentially?


Given the substantial weaknesses and the above questions, my *preliminary* vote for this paper is therefore not to accept it. I am happy to be corrected in any of the criticisms I raised and look forward to the authors' rebuttal.

---

> ### Author Response · Authors · 2023-11-17
>
> We thank the reviewer for their careful reading of our manuscript and the constructive comments. We address below their questions.
>
> > How many dimensions are needed to span $\hat{w}_t$ for all $t$
> ? [...] I conjecture that two dimensions are enough[...]it would make clear that the model only has access to the empirical means
>
> We thank the reviewer for this suggestion. The weight vectors can indeed be completely equivalently rewritten in terms of $\xi,\mu_{emp.} $ where $\mu_{emp.}$ is the empirical means
> $$
> \mu_{emp.}=\frac{1}{n}\sum\limits_{\mu=1}^n s^\mu x_1^\mu=\mu+\frac{1}{n}\eta.
> $$
> Note that this is just an equivalent way to present the equation, and that the underlying results remain totally unchanged. We chose the initial presentation in terms of $\xi, \eta, \mu$ to put further emphasis on the parameters of the target distribution.
> We however agree with the reviewer that using $\{\eta, \mu_{emp.}\}$, rather than $\{\eta,\mu,\xi\}$, constitutes a more concise rephrasing of the results, and have rewritten the manuscript in this way. We have updated the pdf of our paper to reflect this.
>
>
>
>
> > then the model actually does learn the training data by heart (i.e. it predicts the empirical means),
>
> The reviewer is correct that the model overfits, as the trained weights bear explicit dependence on the training samples, and the Gaussian noises $x_0$ employed during training. By "not memorizing", we rather mean that the generated density is not the discrete empirical distribution supported on the training data, but instead a Gaussian mixture. In other words, the generative model allows to generate novel samples, distinct from the training data. We emphasized this distinction in updated manuscript.
>
> On the other hand, it is not entirely correct that the generated density has the empirical mean of the training data as cluster mean. It is already not the case for the Bayes-optimal estimate, which is the empirical means rescaled by a $1/(\sigma^2+n)$ factor. For the generative model, the cluster mean is a linear combination of $\xi$ (subsuming the effect of the noises used during training) and the empirical means, rescaled by multiplicative factors whose expression depend non-trivially on the model parameters - notably the schedule functions $\alpha(t),\beta(t)$, and the regularization $\lambda$. We believe that this precise characterization of the cluster mean of the generated density as a function of the parameters of the learning model is a strength of our analysis.
>
> >Also, the rate $\Theta(1/n)$
>  is not affected by the regularization
> , and the network architecture is not varied so as to judge whether this particular setup has a particular convergence rate. [...]
>
> The main focus of the paper is not to study the dependence of the convergence rate as a function of the model architecture and parameters, but rather show, in a particular setting, how a learning model can manage to learn the target density with a reasonably fast rate only _from a few samples_. This is _allowed_ by the network architecture : indeed, had the minimization (5) been carried out over the space of denoising functions, instead of the parameter space of the considered learning model, the resulting generative model would memorize the training samples, and only allow to generate samples already present in the
> train set, instead of sampling the Gaussian mixture. Furthermore, a discussion for another architecture can already be found in Appendix D of the supplementary material, where we detail the case of an auto-encoder without skip connection. We completely agree with the reviewer that the study of a greater number of architectures is an important research direction to be explored in future works.
>
> >the rate to the true solution is essentially given how fast empirical means converge to the mean of the generative distribution.
>
> This is indeed the correct intuition. We would however like to stress that the mean of the generated density is not equal to the empirical means, and that it is quite non-trivial that an auto-encoder can learn to generate a Gaussian mixture with such means just from a finite number of samples when trained on a denoising loss, and that the learning and transport processes can be sharply theoretically characterized.

---

> > ### Author Response · Authors · 2023-11-17
> >
> > > Why is $\lVert \mu\lVert^2\sim d$
> >  a reasonable scaling? This seems like an unrealistic choice to me.
> >
> >  We thank the reviewer for the relevant question. We would like to draw their attention to the fact that what actually specifies the model is the relative scaling of the norm of the cluster means *and* that of the cluster variances, rather than the former alone. In our model, $\lVert \mu\lVert^2=\Theta(d)$ and the trace of the cluster covariances is $\sigma^2 d=\Theta(d)$. On the other hand, for example for MNIST ($d=784$), taking one class to be one cluster of a mixture, if the data is scaled so that the trace of the covariance is $\approx d$ like in our setting, the cluster means then have squared norm $\lVert \mu\lVert^2\approx 350 $, which is to a large extent of the same order as $d=784$. For FashionMNIST ($d=784$), similarly $\lVert \mu\lVert^2\approx 420 $. While this is of course data-dependent, this scaling is thus reasonable for some simple real settings.
> >
> > > Does the sample complexity also transfer to $\lVert\mu\lVert=\Theta(1)$
> > ? I would guess that the other directions start playing a role then.
> >
> > The reviewer is indeed correct that the phenomenology differs when $\lVert\mu\lVert^2=o(d)$, and indeed the trained weights $\hat{w}_t$ are no longer contained in a low-dimensional subspace at all times. This sizeably complicates the analysis. In particular, when $\lVert\mu\lVert=\Theta(1)$, we expect that $n=\Theta(d)$ training samples are needed for the model to achieve non-trivial performance. One example of such asymptotic regime is $n,d\to \infty$ while $\alpha=n/d=\Theta(1)$. From previous works on learning in such regimes (e.g. (Cui & Zdeborová, 23)), we expect the error to scale as $\Theta(1/\alpha)$. Since the phenomenology of this scaling substantially differs from the one discussed in the present work, we believe a detailed analysis thereof warrants a separate future work.
> >
> > > Can you provide a simple drawing of
> >  and a learnt trajectory? If the required dimension is indeed two, this should be easy.
> >
> >  We have revised Fig. 2 (middle), and included such a trajectory, alongside more discussion in the text. This revised figure can be found in the updated pdf.
> >
> >
> > > What is the shape of the learnt distribution within the two clusters, i.e. what is the local density in each cluster?
> >
> > As described in the paragraph above Corollary 3.3, the distribution the generated data in $\mathrm{span}(\xi,\mu_{emp.})^\perp$ is Gaussian. The variance is given by the last term in (22). The distribution of $X_t$ in $\mathrm{span}(\xi,\mu_{emp.})$, as characterized by the distribution of its two corresponding components $ Q^\xi_t,M_t$, on the other hand concentrates to the values characterized by Result 3.1.
> >
> >
> > >Fig. 2: Why are the learnt means biased in one direction? Is more training data added sequentially?
> >
> > It is a correct observation. In Fig. 2 (right), several setups, corresponding to differing number of samples $n$, are represented in a single plot. For visual clarity of the PCA projection, the training set for an experiment with $n_1$ samples was taken to be also contained in the training set for an experiment with a bigger number of samples $n_2>n_1$. We have emphasized this technicality in the updated manuscript. Note that this choice was made purely for visual clarity purposes and does not bias the individual  experiments or theory in any way.
> >
> > >Minor points
> >
> > We thank the reviewer for identifying these typos, and have fixed them in the updated pdf. We have however chosen to keep the statement of the main contributions as bullet points for better clarity and conciseness.

---

> ### Comment · Reviewer_v6CB · 2023-11-20
>
> I thank the authors for the extensive answers and update of the manuscript.
>
> I appreciate that the authors undertook the effort to show that the main behavior of the flow can be described by the span of $\xi, \mu_\text{emp.}$. This makes clear that the model does not have implicit knowledge of the underlying distributions, but really the convergence is successfully borrowed from the convergence of the empiric mean (and not worse).
>
> **However, my concern remains that the research question is not really addressed.**
>
> Let us maybe approach this starting from my Q4: The characterization of the learned density in Corollary 3.3 describes the unspanned directions as Gaussian, and the distribution in the relevant subspace $\operatorname{span}(\xi, \mu_\text{emp.})$ is specified as time-recursive formulas of the components in $\xi$ and $\mu_\text{emp.}$ directions, influenced by the corresponding components of the weights. What does this tell me about the learned distribution at each of the mixtures? The authors say that the DAE with skip connections is superior over the DAE without skip connections. However, Appendix D shows the same scaling behavior with $n$ is observed in the metric of Corollary 3.3. Even the other end of the spectrum, a fully flexible function memorizing the training data, has this convergence behavior.
>
> So under what metric should architectures be compared, how do the architectures differ in this metric, and what should we conclude from this on the inductive bias & sample complexity of models? In my view, this metric is a strong metric of convergence such as KL divergence or total variation, and the authors are able to show convergence as $\Theta(1/n)$ if this is indeed the dominant scaling behavior.
>
> I am sceptical that the theoretical insight of this work is not large enough for being interesting to a broader audience. Due to the granularity of the scoring system, I think that the improvements are currently not enough to improve my rating. However, I am still happy to be convinced otherwise.

---

> > ### Author Response · Authors · 2023-11-21
> >
> > Reading the reviewer's original questions in view of their last answer, we realize that there may be a misunderstanding that requires clarification. The vector $\mu_{emp.}$ learnt by the DAE (see e.g. equation (12) and our last answer) is the quantity
> >
> > $$
> > \frac{1}{n}\sum\limits_{\mu=1}^n s^\mu x_1^\mu~~~~~~~~~~~~(1)
> > $$
> > It is _not_ the empirical average of the samples (which is on average zero for the Gaussian mixture)
> > $$
> > \frac{1}{n}\sum\limits_{\mu=1}^n x_1^\mu    ~~~~~~~~~~~~~~~~(2)
> > $$
> > in that (1) involves the labels $s^\mu$, which are not available to the model. Therefore, it is not true that  "a fully flexible function memorizing the training data, has this convergence behavior", since the mean of the memorized training distribution is the empirical average (2) which does not converge to the means of the two clusters (their distance is $\Theta_n(1)$). On the other hand, the DAE parametrized model learns a mixture centered around the vector (1) (plus a component along $\xi$) and the opposite thereof, and (1) converges to the true mean $\mu$.
> >
> >
> >
> > >However, my concern remains that the research question is not really addressed.
> >
> > The question of inductive and architectural bias is a motivation of our work, but is not the research question. We have further made this clear in the introduction. The research question and focus of the work is the derivation of the first (to the best of our knowledge)  _sharp_ analysis of a generative flow learnt from limited data. We nevertheless address the reviewer's concerns in detail below.
> >
> > >What does this tell me about the learned distribution at each of the mixtures?
> >
> > The learnt distribution is a bimodal balanced Gaussian mixture centered around $\pm\hat{\mu}$ (characterized in Result 3.1), and with variance
> > $$
> >  \hat{\sigma}^2=e^{2 \int_0^1
> >     dt(
> >     \dot{\beta}(t) \hat{c}(t)+\frac{\dot{\alpha}(t)}{\alpha(t)}(1-\hat{c}(t) \beta(t)))
> >     }
> > $$
> > This is also true in $\mathrm{span}(\xi,\mu_{emp.})$. We have explicited all the details in a new Appendix F, see eq. (118).
> >
> > >In my view, this metric is a strong metric of convergence such as KL divergence or total variation, and the authors are able to show convergence as $1/n$ if this is indeed the dominant scaling behavior.
> >
> > We provide in Appendix F a derivation of a closed-form expression for the Wasserstein distance between the target and generated densities. We employ the mixture Wasserstein distance proposed in [a], which constitutes a natural generalization of the standard Wasserstein distance for Gaussian mixture distributions and further presents the advantage of being analytically tractable. In this metric, we show that the distance between the generated and target density decays as $\Theta_n(1/n)$. We included pointers to this result in the main text.
> >
> > [a] Delon and Desolneux, _A Wasserstein-type distance in the space of Gaussian Mixture Models_, SIAM 2020
> >
> >
> >
> > >The authors say that the DAE with skip connections is superior over the DAE without skip connections. However, Appendix D shows the same scaling behavior with  is observed in the metric of Corollary 3.3. Even the other end of the spectrum, a fully flexible function memorizing the training data, has this convergence behavior.
> >
> > As we show in Appendix F, for a DAE _without_ skip connection, the generated density is bounded away from the target density in Wasserstein distance, with the latter remaining $\Theta_n(1)$ (i.e. it does not improve with the number of samples). Similarly, a fully flexible function memorizing the data also leads to a $\Theta_n(1)$ Wasserstein distance, and therefore _fails to sample from the target distribution_. We have further included an Appendix G where we make analytically explicit in a closely related setup how a neural network with infinite expressivity memorizes the training data.

---

> > > ### Comment · Reviewer_v6CB · 2023-11-22
> > >
> > > I thank the authors for the additional information that provides additional insight.
> > >
> > > The additional and explicit characterization of the learned density is quite insightful, as it explicitly describes what the distribution is. I think that this discussion is very useful: For a paper targeting a sharp analysis of the learned distribution, I would expect an easily interpretable functional form, which seems easy to derive according to the authors.
> > >
> > > However, I am not convinced about their derivation: To obtain the shape of the learned mixture components, the authors derive the first order correction term of a sample from x_0=0 if I understand correctly. Thus, the derivation only computes the second moment of a mixture component. This does not show that the learned density is actually Gaussian. It is encouraging to see that the standard deviations seem to converge to the correct value.
> > >
> > > In this light, I am surprised to see that the DAE without skip connections learns a $\delta$ distribution. What happens if you apply the same argument as for the DAE with skip connections, i.e. what standard deviation is learned?
> > >
> > > Regarding the provided mixture Wasserstein distance, it does not seem to be a generalization of the Wasserstein distance, but rather a related concept ("Wasserstein-type distance" as per the referenced paper). In particular, it only applies to Gaussian mixtures. In light of the above argument, it seems that the learned density does not obtain Gaussian shape and therefore the mixture Wasserstein distance does not apply.
> > >
> > > Are the above points valid?

---

> > > > ### Author Response · Authors · 2023-11-23
> > > >
> > > > We understand that the reviewer now agrees with the asymptotic results presented in the main text and that the research question has been clarified. Their present questions bear on the new Appendix F that stemmed from a suggestion they made two days ago, which we found interesting, but in our view, the contributions of our paper are valuable also without it.
> > > >
> > > > >In this light, I am surprised to see that the DAE without skip connections learns a
> > > >  distribution. What happens if you apply the same argument as for the DAE with skip connections, i.e. what standard deviation is learned?
> > > >
> > > > Using the same argument when the skip connection is $c_t=0$, the variance is given by
> > > > $$
> > > > e^{2\int_0^1\dot{\alpha}(t)/\alpha(t)dt}=(\alpha(1)/\alpha(0))^2=0
> > > > $$
> > > > Thus the skipped connection is needed for the performance of the model to be good.
> > > >
> > > > >In particular, it only applies to Gaussian mixtures.
> > > >
> > > > Note that in the way the Mixture Wasserstein is defined, it can be applied to any mixture distributions, and not necessarily Gaussian ones.
> > > >
> > > > Please also note that the cluster distribution in a dimension $2$ subspace is asymptotically (in the large $d$ limit considered in our paper) irrelevant for all considered metrics (including the Mixture Wasserstein). We show that the generated distribution is Gaussian in the $d-2$ directions and thus is the $d\to\infty$ limit, it is thus Gaussian in the leading order.

---

### Author Response · Authors · 2023-11-17

We thank all the reviewers for their insightful comments. We answer each of them in detail below. Following the reviewers' input, we updated the manuscript, where we rephrased the presentation of our results and figures for enhanced clarity, and included additional discussions.

---

### Meta-Review · Area_Chair_irJp · 2023-12-07

**Metareview:**

All reviewers agree that this work provides the first sharp asymptotic analysis of a generative flow in an idealized setting, which in turn gives the exact characterization of all quantities. The AC thus recommends acceptance.
The authors are encouraged to discuss more limitations of this work according to the reviews.

**Justification For Why Not Higher Score:**

This paper only studies a highly idealized setting. There is still a gap between the model studied and the practical models.

**Justification For Why Not Lower Score:**

This work gives the first sharp asymptotic analysis of a generative flow.

---

### Decision · Program_Chairs · 2024-01-16

Accept (poster)